# Light-based tuning of ligand half-life supports kinetic proofreading model of T cell signaling

Doug K Tischer[1,2], Orion David Weiner[1]*

[1]Cardiovascular Research Institute, University of California, San Francisco, San Francisco, United States; [2]Department of Biochemistry and Biophysics, University of California, San Francisco, San Francisco, United States

**Abstract** T cells are thought to discriminate self from foreign peptides by converting small differences in ligand binding half-life into large changes in cell signaling. Such a kinetic proofreading model has been difficult to test directly, as existing methods of altering ligand binding half-life also change other potentially important biophysical parameters, most notably the mechanical stability of the receptor-ligand interaction. Here we develop an optogenetic approach to specifically tune the binding half-life of a chimeric antigen receptor without changing other binding parameters and provide direct evidence of kinetic proofreading in T cell signaling. This half-life discrimination is executed in the proximal signaling pathway, downstream of ZAP70 recruitment and upstream of diacylglycerol accumulation. Our methods represent a general tool for temporal and spatial control of T cell signaling and extend the reach of optogenetics to probe pathways where the individual molecular kinetics, rather than the ensemble average, gates downstream signaling.
DOI: https://doi.org/10.7554/eLife.42498.001

*For correspondence:
orion.weiner@ucsf.edu

Competing interests: The authors declare that no competing interests exist.

## Introduction

The T cell response begins when the T cell antigen receptor (TCR) binds its cognate peptide-major histocompatibility complex (pMHC) on an antigen-presenting cell. Ligand discrimination is essential, as falsely recognizing a self-peptide can lead to autoimmunity, while missing a foreign-peptide allows pathogens to evade detection. Compounding this challenge, self-pMHCs generally outnumber foreign-pMHCs by several orders of magnitude (*Unternaehrer et al., 2007*; *Cohen et al., 2003*; *Bhardwaj et al., 1993*; *Irvine et al., 2002*). How the TCR recognizes its foreign-pMHC while ignoring the vastly more numerous self-pMHC is a fundamental open question in immunology.

For many cell surface receptors, occupancy drives the strength of signaling. Lower affinity ligands can signal as well as high affinity ligands when their concentration is increased such that an equal number of receptors are bound (*Kinzer-Ursem et al., 2006*). This is unlikely to be the case for activation of the T cell, because low affinity pMHCs are significantly less potent than high affinity pMHCs, even when adjusted for occupancy (*Daniels et al., 2006*). Furthermore, given the large excess of self-pMHC (*Unternaehrer et al., 2007*; *Cohen et al., 2003*; *Bhardwaj et al., 1993*; *Irvine et al., 2002*) and the relatively small differences in binding affinity (*Davis et al., 1998*; *Germain and Stefanová, 1999*; *Gascoigne et al., 2001*), it is likely that more TCRs are bound to self-pMHC than foreign-pMHC. For a T cell to detect foreign pMHCs, it is likely that some ligand-intrinsic factor makes bound foreign-pMHCs more stimulatory than bound self-pMHCs.

A major model of T cell ligand discrimination is kinetic proofreading (*McKeithan, 1995*), which predicts that small differences in ligand binding half-life can be amplified into large differences in signaling. Such a model is attractive because it allows a few receptors bound to long-lived ligands to

signal better than many receptors bound to short-lived ligands, potentially explaining why abundant self-pMHCs do not activate T cells, though scarce foreign-pMHCs do. It posits a delay made up of multiple irreversible biochemical steps between ligand binding and downstream signaling. Only ligands that bind persistently signal effectively. These steps only occur while the ligand is bound and quickly reset upon dissociation, preventing two successive short binding events from mimicking one long binding event. This mechanism is based on the kinetic proofreading that gives DNA replication, protein translation (*Ninio, 1975*; *Hopfield, 1974*), and mRNA splicing (*Burgess and Guthrie, 1993*) specificity far beyond what would be predicted from equilibrium binding constants alone. Consistent with this hypothesis, kinetic measurements generally show that stimulatory pMHCs have longer binding half-lives, while non-stimulatory pMHCs have shorter binding half-lives (*Davis et al., 1998*; *Germain and Stefanová, 1999*; *Gascoigne et al., 2001*; *Matsui et al., 1994*).

Kinetic proofreading effects need to be strongest when differences in pMHC binding affinities are small, as abundant self-pMHCs will bind more TCRs than scarce foreign-pMHCs. Strong kinetic proofreading ensures that short self-pMHC binding events are unlikely to contribute to downstream signaling. However, recent techniques that measure pMHC binding affinities in more native 2D environments show a larger range of binding affinities than previously measured (*Huang et al., 2010*), suggesting that there may not be as large an excess of self-pMHC bound to the TCR and raises the question of the magnitude of kinetic proofreading needed to discriminate pMHC ligands.

To probe kinetic proofreading in T cells, the field most commonly relies on altered peptide series to change binding half-lives. The limitation of this approach is that when the binding interface is altered, both the bond's half-life and stability under tension must change together (*Figure 1A*, left). Differentiating the effects of these two variables is essential because recent evidence has shown that the TCR is mechanosenstive (*Kim et al., 2009*; *Feng et al., 2017*; *Das et al., 2015*) and that foreign-pMHC can form catch bonds with the TCR, growing more stable under load (*Liu et al., 2014*). Thus, conventional mutational approaches make it difficult to determine whether changes in signaling arise from alterations in binding half-life or reflect changes in the force being applied to the TCR. To specifically probe the role of binding half-life in ligand discrimination, the field needs new techniques to manipulate the receptor-ligand half-life while leaving force transmission and all other aspects of the interaction unchanged. Here we develop an optogenetic approach for T cell activation that addresses this need.

We chose the LOVTRAP system for optogenetic control of bind half-life because it is one of the few systems where light stimulates the dissociation of two proteins on the order of seconds (*Ni et al., 1999*; *Wang et al., 2016*), the approximate physiologic range of pMHC-TCR interactions. We sought to utilize LOVTRAP's ability to disrupt a protein-protein interaction as a way of controlling the binding half-life of a synthetic ligand-receptor pair.

LOVTRAP consists of two parts: a blue-light sensitive protein (LOV2) and an engineered binding partner (Zdk). Zdk preferentially binds LOV2 in the dark but quickly unbinds when a photon of blue light excites LOV2, inducing a conformational change. Since photons of blue light stimulate Zdk dissociation, low photon fluxes enforce longer binding half-lives while high photon fluxes enforce shorter binding half-lives. The intrinsic kinetics of Zdk dissociation from the ground and excited states of LOV2 set the upper and lower bounds on the range of achievable half-lives (approximately 500 ms to 10 s in our hands; *Figure 2B*).

To construct our synthetic ligand-receptor pair, we used a chimeric antigen receptor (CAR) approach, which has the advantage of being simpler than the TCR, easier to engineer, and medically relevant in its own right (*Barrett et al., 2014*; *Sadelain et al., 2013*). CARs minimally consist of an extracellular ligand binding domain, a transmembrane domain and an intracellular signaling domain. Because CARs are not pMHC restricted, Zdk served as the extracellular ligand binding domain. Biochemically purified LOV2 presented on a supported lipid bilayer (SLB) served as the ligand while allowing for simultaneous imaging of live cell signaling reporters.

A key advantage of this approach is that it allows us to test the kinetic proofreading model directly. For most protein-protein interactions, the binding half-life and the bond's stability under tension necessarily co-vary with each other, as they are both consequences of the height of the transition state. This makes it hard to distinguish between a kinetic proofreading and mechanosensitive model of T cell signaling. A ligand with a long binding half-life is *necessarily* more stable under load, and both models predict it would be more stimulatory. Our approach uncouples these parameters by using one ligand-receptor pair to explore a range of half-lives. Blue light, not point mutations,

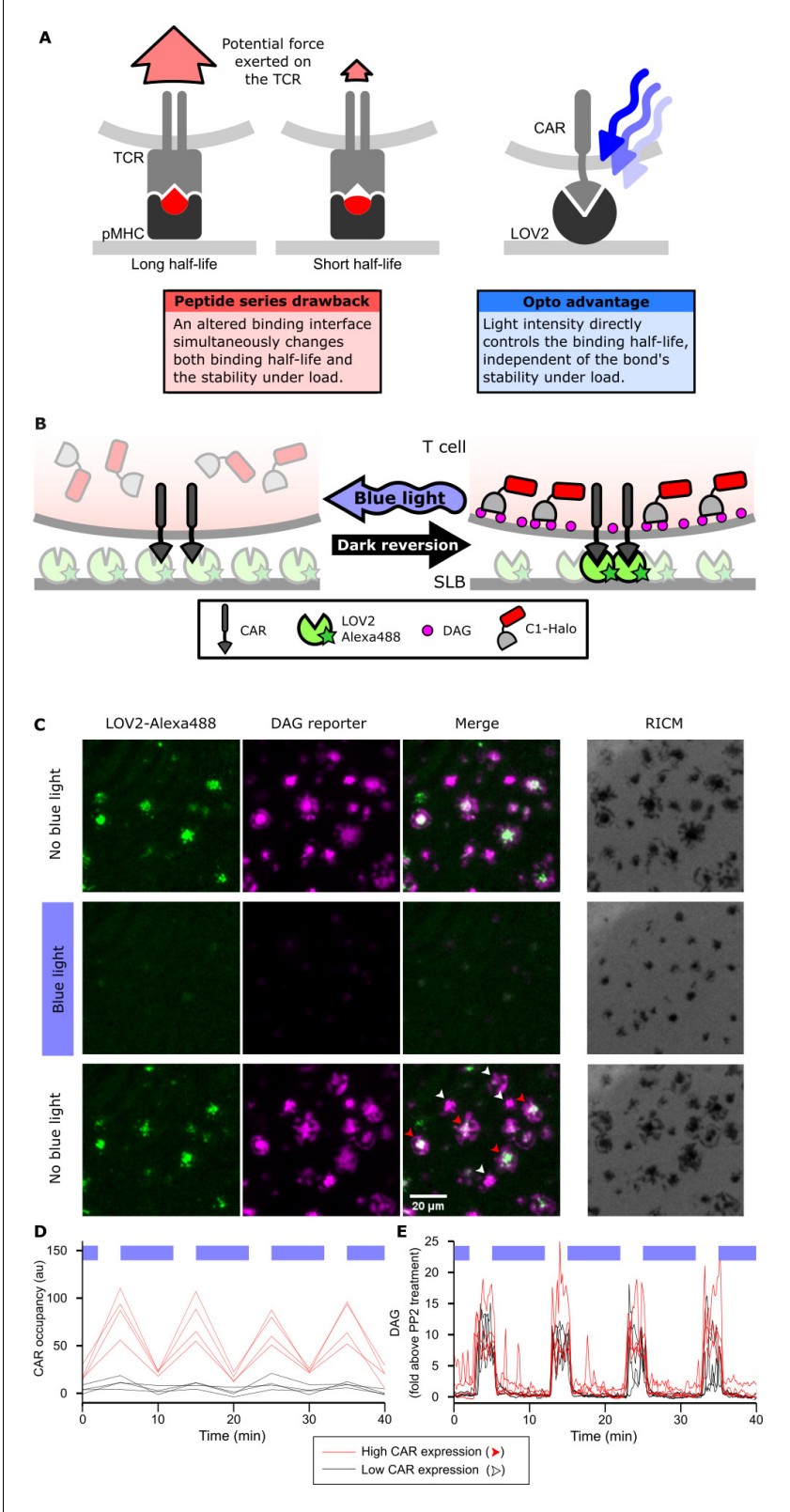

**Figure 1.** Strategy for testing kinetic proofreading with optogenetic tools. (**A**) Conventional methods of mutating the pMHC to alter the binding half-life also change the binding interface, which changes several parameters at once. By contrast, optogenetic control allows light intensity to control ligand binding half-life while keeping the binding interface constant. Therefore, no other aspects of the receptor-ligand interaction change. Red and blue

*Figure 1 continued on next page*

*Figure 1 continued*

lines highlight the binding interfaces. (B) Schematic of experimental setup. Jurkat cells expressing a live cell DAG reporter and a Zdk-CAR are exposed to an SLB functionalized with purified, dye-labeled LOV2. In the dark, LOV2 binds to and accumulates under the receptor, and stimulates DAG production, recruiting the reporter to the plasma membrane. Blue light excites LOV2, inducing its dissociation from the receptor to terminate signaling. (C) Montage from a time course in which cells were alternately stimulated in the presence or absence of blue-light. White arrows highlight cells with low to undetectable receptor occupancy, and red arrows highlight cells with high receptor occupancy. (D and E) Cells with very different levels of receptor occupancy (D), can have similar DAG levels (E), suggesting that receptor occupancy is not a good predictor of DAG levels. Top blue bars indicate the presence of blue light.

DOI: https://doi.org/10.7554/eLife.42498.002

The following figure supplements are available for figure 1:

**Figure supplement 1.** Light-based control of T cell signaling is durable for hours.
DOI: https://doi.org/10.7554/eLife.42498.003
**Figure supplement 2.** Cells spread in response to blue-light illumination.
DOI: https://doi.org/10.7554/eLife.42498.004
**Figure supplement 3.** Colocalization of ligand binding and downstream signaling.
DOI: https://doi.org/10.7554/eLife.42498.005
**Figure supplement 4.** Absolute quantification of cell surface TCRs and CARs.
DOI: https://doi.org/10.7554/eLife.42498.006

tunes the binding half-life. Because the ligand-receptor pair remains constant in all experiments, so too does the amount of tension they can withstand. Our optogenetic approach directly and specifically tunes ligand binding half-life, allowing us to cleanly measure the degree to which binding half-life influences T cell signaling.

A point of controversy is whether kinetic proofreading steps occur at the TCR (*Taylor et al., 2017*; *Stepanek et al., 2014*; *Mandl et al., 2013*; *Sloan-Lancaster et al., 1994*; *Madrenas et al., 1997*) or further downstream (*O'Donoghue et al., 2013*). An advantage of our synthetic CAR approach is that it is simpler than the TCR, helping to bypass some early signaling steps (e.g. CD4 or CD8 coreceptor involvement which are lacking in the CAR; *Harris and Kranz, 2016*) and focus on the role the shared downstream pathway can play in ligand discrimination. Paired with live cell read-out at multiple steps in the signaling pathway, our approach helps to define the degree to which different portions of the pathway contribute to kinetic proofreading.

By directly controlling ligand binding half-life with light and holding all other binding parameters constant, we show that longer binding lifetimes are a key parameter for potent T cell signaling. Surprisingly, this discrimination occurs in the proximal signaling pathway, downstream of ZAP70 recruitment and upstream of DAG accumulation. This work aids our understanding of how T cell discriminate ligands and expands optogenetics as a tool for controlling the timing of single molecular interactions.

## Results

### LOV2 photoreversibly binds the CAR

We first validated the ability of the LOV2 ligand to photoreversibly bind the Zdk-CAR. Clonal Jurkat cells stably expressing the Zdk-CAR were exposed to SLBs functionalized with purified Alexa-488-labeled LOV2 (*Figure 1B*). Because LOV2 diffuses freely in the bilayer and becomes trapped upon interaction with the Zdk-CAR, we can measure receptor occupancy by the accumulation of LOV2 under the cell. As expected, LOV2 accumulated under the cells in the absence of blue light and dispersed following illumination with blue light (*Figure 1C*, *Video 1* and *2*). Blue light drives multiple cycles of binding and unbinding without apparent loss of potency (*Figure 1D* and *Figure 1—figure supplement 1A*).

### LOV2 binding induces cell signaling

We next verified that binding of the LOV2 ligand induces cell signaling through the CAR. We measured accumulation of the lipid diacylglycerol (DAG) by quantifying the translocation of the C1-

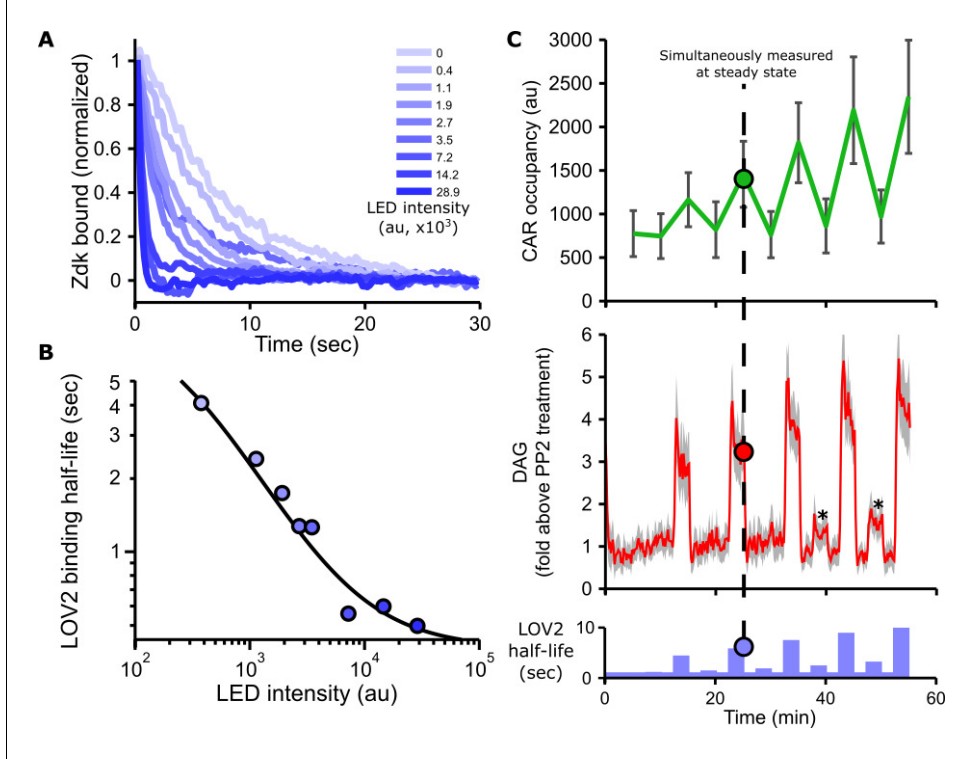

**Figure 2.** Blue light intensity titrates binding half-life, CAR occupancy and DAG levels. (**A**) In vitro measurements of blue light intensity-based control of LOV2-Zdk binding half-life. SLBs functionalized with LOV2 were combined with soluble, dye-labeled Zdk. After washing out free Zdk, Zdk dissociation was measured upon acute illumination with different intensities of blue light using TIRF microscopy. (**B**) Blue light intensity enforces LOV2-Zdk binding half-lives from ten seconds to hundreds of milliseconds. Binding half-lives were determined by fitting a single exponential decay to the traces shown in **A**). Data was fit using a two-step unbinding model (solid black line), consisting of light-dependent excitation of LOV2 followed by light-independent release of Zdk. (**C**) Time course showing that intermediate light levels modulate ligand binding half-lives (bottom), titrate receptor occupancy (top), and induce DAG accumulation in Jurkat cells (middle). Asterisks in middle panel highlight small but detectable increases in DAG levels to weak stimuli. n = 31 cells. Mean with 95% CI (two-sided Student's t-test).
DOI: https://doi.org/10.7554/eLife.42498.007

The following figure supplements are available for figure 2:

**Figure supplement 1.** Purification of LOV2 and Zdk1.
DOI: https://doi.org/10.7554/eLife.42498.008
**Figure supplement 2.** Calculating DAG levels.
DOI: https://doi.org/10.7554/eLife.42498.009
**Figure supplement 3.** Calculating CAR occupancy.
DOI: https://doi.org/10.7554/eLife.42498.010

domains from protein kinase C theta from the cytosol to the plasma membrane with Total Internal Reflection Fluorescence (TIRF) microscopy (**Figure 1B**). DAG levels spiked and cells spread onto the SLBs in a blue-light-gated fashion (**Figure 1C,E** and **Video 3**).

We observed no blue-light-dependent changes in DAG levels when LOV2 was omitted from the bilayer (**Figure 1—figure supplement 1C**). Additionally, a ZAP70-mCherry reporter co-localized well to sites of CAR occupancy (**Figure 1—figure supplement 3**). Although our ligand density was too high to resolve single molecule CAR-LOV2 interactions, ZAP70 was effectively recruited to larger clusters of occupied CARs, confirming that our blue-light-dependent signaling originated from the CAR. DAG showed roughly diffuse localization throughout the cell, consistent with its diffusion away from sites of production (**Figure 1—figure supplement 3**).

Not only the localization, but also the speed of intracellular signaling is consistent with previous measurements. Stimulating T cells with a photoactivatable pMHC shows an ~8 s 'offset' time to

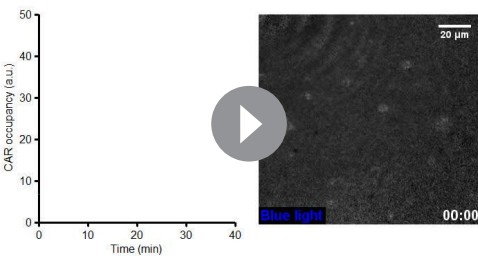

**Video 1.** LOV2 reversibly binds the CAR. Time course showing the photoreversible binding of LOV2-Alexa488 to cells expressing the Zdk-CAR in the presence or absence of strong blue light. Images taken in TIRF with a 488 nm laser. Because the 488 nm laser strongly activates LOV2, LOV2 localization can only be imaged at the end of a pulse of blue light (every five minutes). The mean response of the cells is plotted.
DOI: https://doi.org/10.7554/eLife.42498.011

increased DAG levels. While the slow dark reversion of LOV2 prevents true acute stimulation of our cells, both CAR occupancy and DAG levels rise around 30 s after removing blue light (*Video 2*, *Video 3*), consistent with DAG production beginning within seconds of receptor engagement.

These data confirm that the photoreversible binding of LOV2 to the Zdk-CAR leads to photoreversible signaling. Low DAG levels in the presence of blue light was not due to cells detaching from the SLB, as cells were passively adhered with an anti-β2 microglobulin antibody (*Video 4*), and Reflection Interference Contrast Microscopy (RICM) indicated that the cells remained in continuous contact with the SLB (*Figure 1C*, *Figure 1—figure supplement 2* and *Video 5*).

The CAR was expressed at between 62–238 $\times$ $10^3$ molecules on the cell surface, which is slightly lower but comparable to the native TCR complex in Jurkats (273–547 $\times$ $10^3$ molecules; *Figure 1— figure supplement 4*). Cells were sensitive to between ~40–1,200 LOV2 molecules/um$^2$ on the bilayer (*Figure 4—figure supplement 2* and Methods), which is comparable in sensitivity to other CARs with precisely tunable ligands (*Taylor et al., 2017*) and slightly less sensitive than primary T cell presented with pMHCs on SLBs (*Manz et al., 2011*).

We noticed that within the same field of view, cells could have high or low-to-undetectable receptor occupancies (red and white arrows, respectively; *Figure 1C,D*). However, despite clear differences in receptor occupancy, all cells had similar DAG levels when exposed to similar doses of blue light (*Figure 1E*). This result suggested that something other than receptor occupancy dominates downstream signaling.

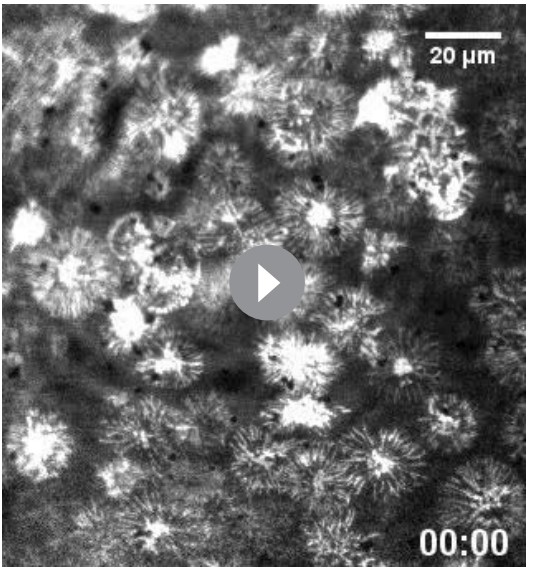

**Video 2.** LOV2 quickly unbinds the CAR. Time course showing that LOV2 binds and unbinds the CAR on the order of seconds in response to blue light. To track LOV2 binding in real time, LOV2 was labeled with Cy3 (instead of Alexa 488 which is used in the rest of the paper) and imaged in TIRF with a 561 nm laser. Such real-time imaging of LOV2 localization could not be used throughout the paper, as the 561 nm laser used to image the DAG reporter was the only laser on the microscope that did not activate LOV2. The dark holes are imperfections in the supported lipid bilayer.
DOI: https://doi.org/10.7554/eLife.42498.012

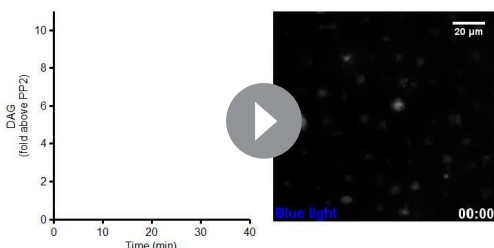

**Video 3.** DAG signaling is photoreversible. Time course showing optogenetic control CAR signaling, as measured by the photoreversible accumulation of DAG in the presence or absence of strong blue light. Cells are the same as in *Video 1*, imaged in TIRF with a 561 nm laser. The mean response of the cells is plotted.
DOI: https://doi.org/10.7554/eLife.42498.013

### Blue light intensity titrates Ligand binding half-life, CAR occupancy and DAG levels

As blue light affects the LOV2-Zdk binding half-life, we sought to measure this relation in more detail. The release of purified Zdk from SLBs functionalized with LOV2 was measured upon acute exposure to different blue light intensities (*Figure 2A* and *Figure 2—figure supplement 1*). Increasing blue-light intensities increased Zdk dissociation and tuned the effective binding half-life from ten seconds to approximately 500 ms (*Figure 2B*).

In addition to providing control over binding half-life, manipulating the blue light levels also finely tuned DAG levels and receptor occupancy (*Figure 2C*, *Figure 2—figure supplement 2*, *Figure 2—figure supplement 3*, *Video 6* and *Video 7*). These data show our optogenetic input is highly titratable and directly controls ligand binding half-life.

### Longer ligand binding half-lives drive higher DAG levels, despite equal CAR occupancy

Because increasing ligand binding half-life also increases receptor occupancy, it is difficult to separate the involvement of occupancy and half-life on downstream signaling from a single time course under one set of conditions. However, the influences of half-life and occupancy can be decoupled by keeping the sequence of blue-light stimulation the same but varying the absolute concentration of the LOV2 ligand on the SLB. One cell exposed to a low concentration of LOV2 and a long binding half-life (via low intensity blue light) can have the same receptor occupancy as another cell exposed to a high concentration of LOV2 and a short binding half-life (via higher intensity blue light; *Figure 3A*). Comparing cell signaling in this context is a very sensitive way to detect the effects of binding half-life alone, as receptor occupancy and all other ligand-intrinsic factors (including the bond's rupture force) are held constant.

Conducting multiple experiments with different LOV2 concentrations and gating the data over a narrow range of receptor occupancy shows a clear result: increasing ligand binding half-life increases DAG levels, despite cells having near identical receptor occupancy (*Figure 3B,C* and *Figure 3—figure supplement 1*). Intriguingly, signaling increases the most for binding half-lives between 4–7 s, in close agreement with previous estimates of the binding half-life threshold for stimulatory versus non-stimulatory pMHCs (*O'Donoghue et al., 2013*; *Palmer and Naeher, 2009*; *Huppa et al., 2010*).

Previous work has shown that fast rebinding can also make ligands stimulatory by extending the effective engagement time of the receptor (*Aleksic et al., 2010*; *Govern et al., 2010*). Interestingly, 2D kinetic measurements show much wider ranges of on-rates than off-rates in the OT-I system (*Huang et al., 2010*). Thus, it is important to remember that the lifetime of ligand binding can differ from the effective lifetime of receptor engagement. However, we anticipate the effects of ligand rebinding to be

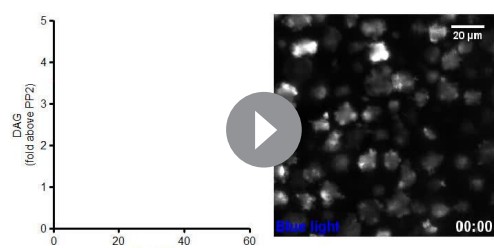

**Video 4.** DAG signaling is LOV2-dependent. Time course showing that in the absence of LOV2, the cells fail to exhibit blue light-induced changes in DAG accumulation. Jurkat cells expressing both the Zdk-CAR and DAG reporter were passively adhered to an SLB lacking LOV2 with a biotinylated anti-β2 microglobulin antibody and were exposed to the same pulses of blue light as in *Video 1*. We observed spontaneous flashes of DAG but no changes in DAG accumulation in response to blue light illumination. Cells were imaged in TIRF with a 561 nm laser. The mean response of the cells is plotted.
DOI: https://doi.org/10.7554/eLife.42498.014

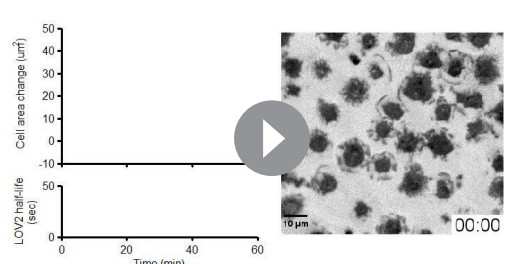

**Video 5.** Light-based titration of cell spreading. Time course showing RICM images of cells exposed to intermediate blue-light intensities. The mean response of cell spreading is plotted.
DOI: https://doi.org/10.7554/eLife.42498.015

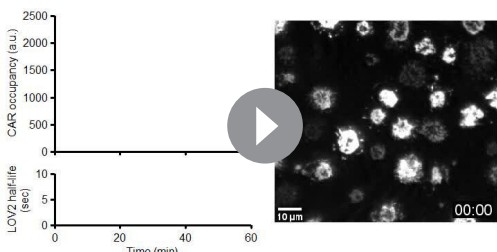

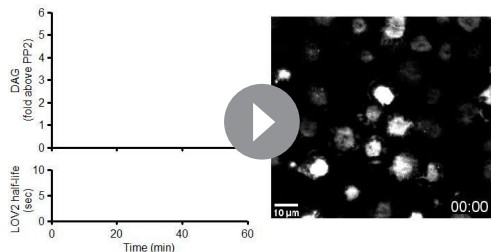

**Video 6.** Light-based titration of CAR occupancy. Time course showing the quantification of receptor occupancy (via LOV2 localization) in response to intermediate blue-light intensities. Because the 488 nm laser strongly activates LOV2, LOV2 localization can only be imaged at the end of a pulse of blue light (every five minutes). Cells are the same as in *Video 5*, imaged in TIRF with a 488 nm laser. The mean response of the cells is plotted.
DOI: https://doi.org/10.7554/eLife.42498.016

**Video 7.** Light-based titration of DAG signaling. Time course showing the quantification of DAG levels in response to intermediate blue-light intensities. Cells are the same as in *Video 5*, imaged in TIRF with a 561 nm laser. The mean response of the cells is plotted.
DOI: https://doi.org/10.7554/eLife.42498.017

reduced in our approach compared with an altered peptide series, as LOV2 is refractory to CAR binding after being stimulated with blue light. Nevertheless, our 4–7 s window separating stimulatory and non-stimulatory half-lives could be an underestimate if our CAR can quickly rebind a molecule of LOV2.

It is important to stress that these binding half-lives are enforced by constant, not periodic, blue light illumination. Mechanisms other than kinetic proofreading could explain reduced signaling if the cells were stimulated in short, periodic bursts. By contrast, constant blue light illumination allows cells to reach a steady state, which is when we make our single cell measurements. Although on the whole receptor occupancy and downstream signaling are constant, individual receptor-ligand pairs are continually forming and dissociating with a half-life set by the intensity of blue light. Under these conditions, differences in signaling are only attributable to differences in the single molecule interactions between receptor and ligand, not global state changes of the cell.

## DAG levels correlate more strongly with ligand binding half-life than CAR occupancy

Stimulating cells with the same intensities of blue light but different concentrations of LOV2 enabled us to measure the effects of binding half-life in the context of low, medium, and high receptor occupancy. Strikingly, the enforced ligand binding half-life predicted DAG levels well, regardless of the amount of ligand present. When DAG is plotted as a function of binding half-life, the response curves from different experiments nearly overlap with each other, showing a strong correlation coefficient (*Figure 3D,E* and *Figure 3—figure supplement 2*, top). By contrast, plotting DAG as a function of receptor occupancy shows a poor correlation (*Figure 3D* and *Figure 3—figure supplement 2*, bottom). Because half-life is a strong predictor of DAG levels across a range of receptor occupancies, these data argue that binding half-life is a major determinant of CAR signaling.

## Quantifying the degree of kinetic proofreading

Combining the data from all experiments over a range of LOV2 ligand concentrations, we sought to quantify how strongly binding half-life influences CAR signaling. In kinetic proofreading, the delay between ligand binding and downstream signaling is modeled as a series of discrete steps (*McKeithan, 1995*). The stronger the proofreading, the greater the number of steps and the more binding half-life dominates downstream signaling. To provide a conservative, lower estimate of the degree of proofreading, we modeled the DAG response as a saturable system downstream of 'strong' proofreading steps, that is the forward biochemical reaction is sufficiently slow that the probability of completing a step is proportional to the ligand binding half-life. In such a model,

$$DAG \propto \frac{R\tau^n}{K + R\tau^n} + \beta$$

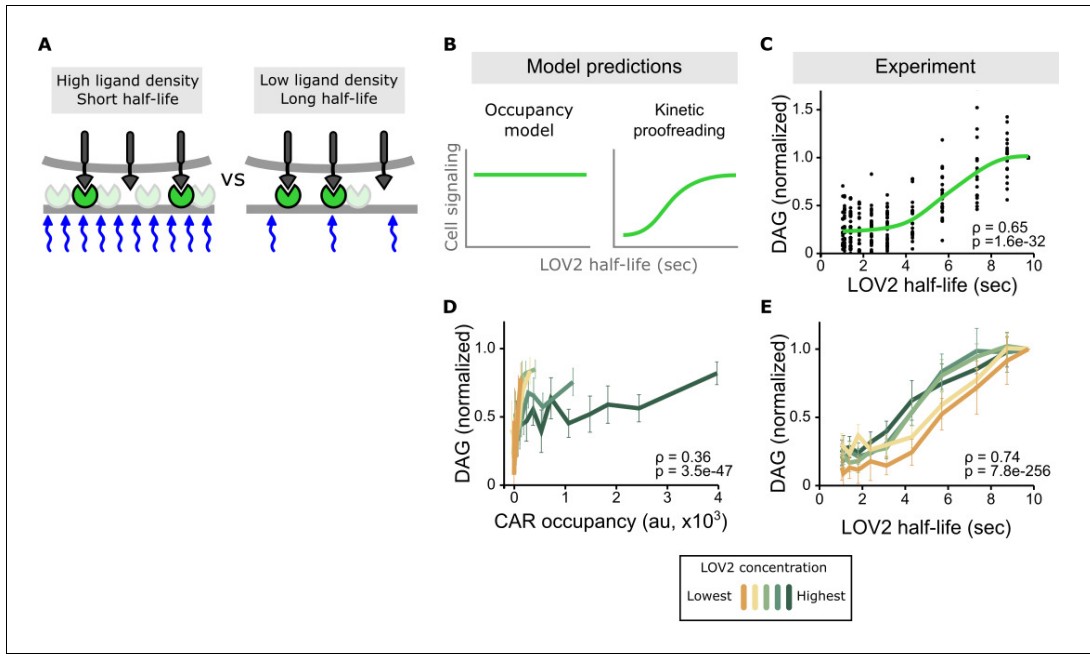

**Figure 3.** Binding half-life, not receptor occupancy, dominates CAR signaling. (**A**) A cell exposed to a high LOV2 density but a short binding half-life can have the same receptor occupancy as a cell exposed to a low LOV2 density but a long binding half-life. (**B**) At constant receptor occupancy, an occupancy model predicts binding half-life should have no effect on signaling, while kinetic proofreading predicts that increasing binding half-life should increase signaling. (**C**) At constant receptor occupancy, increasing ligand binding half-life increases DAG signaling, as shown by both non-parametric kernel smoothing regression (green line) and Spearman's correlation coefficient ($\rho$). Each black dot is a single cell measurement obtained from multiple experiments over a range of LOV2 concentrations. (**D** and **E**) During individual experiments, there is a fixed concentration of LOV2, meaning that both receptor occupancy and binding half-life change in response to blue-light. We measured Spearman's correlation coefficient to ask whether DAG levels were best described as a function of receptor occupancy or binding half-life across different LOV2 concentrations. DAG levels are more correlated with binding half-life (**E**) than they are with CAR occupancy (**D**). This is reflected in the fact that the DAG response curves nearly overlap with each other when plotted as a function of binding half-life, indicating changing the concentration of LOV2 has little effect on downstream signaling. These data are consistent with binding half-life, not receptor occupancy, dominating CAR signaling. The mean is plotted with a 95% CI (two-sided Student's t-test).

DOI: https://doi.org/10.7554/eLife.42498.018

The following figure supplements are available for figure 3:

**Figure supplement 1.** Long LOV2 binding half-lives signal better than short binding half-lives, even at equal receptor occupancy.

DOI: https://doi.org/10.7554/eLife.42498.019

**Figure supplement 2.** DAG levels are most strongly correlated with ligand binding half-life.

DOI: https://doi.org/10.7554/eLife.42498.020

where $R$ is the receptor occupancy, $\tau$ is the enforced ligand binding half-life, $n$ is the degree of proofreading, $K$ is the amount of upstream signal that generates a half-maximal DAG response, and $\beta$ is basal signaling through the pathway.

The value of $n$ measures how strongly ligand binding half-life affects signaling. For $n = 0$, signaling depends only on receptor occupancy and is unaffected by binding half-life. For $n > 0$, there is some degree of kinetic proofreading. As $n$ increases, ligand binding half-life has a larger and larger impact on downstream signaling, to the point that short lived ligands cannot generate as much signal as long-lived ligand simply by increasing concentration and receptor occupancy (*Figure 4A*, dotted line). This aligns with what has been observed with T cells: abundant self-pMHCs do not activate T cells, while a few foreign-pMHCs can activate T cells (*Irvine et al., 2002*; *Kimachi et al., 1997*; *Christinck et al., 1991*; *Demotz et al., 1990*; *Sykulev et al., 1996*).

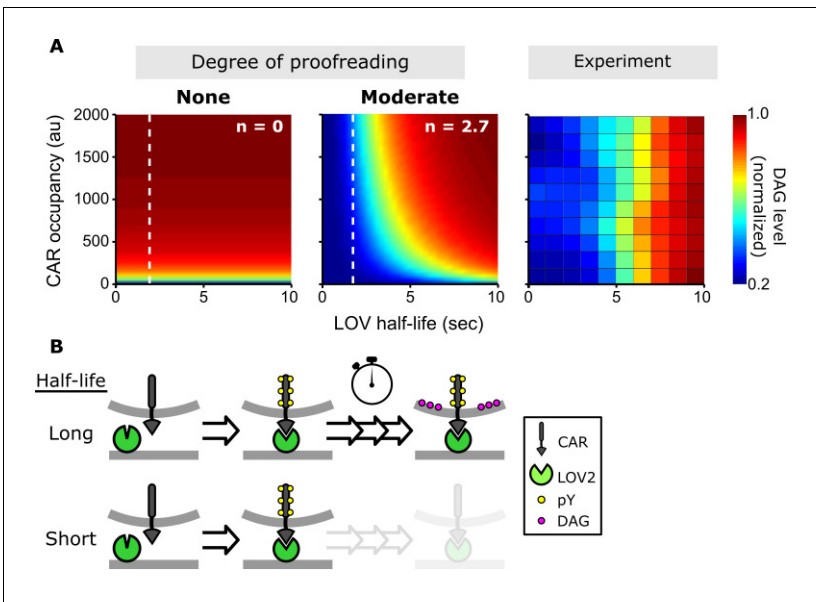

**Figure 4.** A kinetic proofreading model best explains T cell signaling. (**A**) Models for how CAR occupancy and binding half-life affect T cell signaling in the presence of moderate (left) or no kinetic proofreading (middle). To facilitate visualization, single cell measurements were fit with non-parametric kernel smoothing regression and plotted as a heat map (right). The degree of proofreading is denoted by *n*, and the value of *n* for the moderate proofreading scenario is derived from single cell measurements from all three data sets. (*Figure 4—figure supplement 1*). Experimental data of DAG levels as a function of CAR occupancy and binding half-life (right) is consistent with moderate kinetic proofreading. (**B**) Schematic of our kinetic proofreading model. After a ligand binds the receptor, it must remain bound sufficiently long to accommodate the slow proofreading steps. Long lived ligands survive this slow waiting period and produce strong downstream signaling, while short lived ligands dissociate and produce weak downstream signaling.
DOI: https://doi.org/10.7554/eLife.42498.021

The following source data and figure supplements are available for figure 4:

**Source data 1.** This spreadsheet contains all the single cell data used in this study.
DOI: https://doi.org/10.7554/eLife.42498.025
**Figure supplement 1.** DAG normalization method only has a minor effect on the calculated degree of proofreading.
DOI: https://doi.org/10.7554/eLife.42498.022
**Figure supplement 2.** Justification for DAG normalization.
DOI: https://doi.org/10.7554/eLife.42498.023
**Figure supplement 3.** ZAP70 recruitment does not show evidence of kinetic proofreading.
DOI: https://doi.org/10.7554/eLife.42498.024

Fitting our data, we find $n = 2.7 \pm 0.5$ (95% CI; *Figure 4—figure supplement 1* and *2*) which means this degree of proofreading could allow T cells to amplify a 10-fold difference in ligand binding half-life into a ~ 5,000 fold difference in DAG levels. While this result is only true when the signaling pathway is far from saturation, it is reasonable that T cells operate far from saturation when they encounter foreign-pMHCs, given the low stimulatory nature of self-pMHCs. However, how a T cell responds is ultimately a combination of the magnitude of signaling, and the particular set of signaling proteins expressed in different stages of T cell development and in different T cell subsets (*Hogquist et al., 1994*; *Lucas et al., 1999*; *Hogquist et al., 1997*). Ligand kinetics may be more or less important at evoking responses in these different situations.

We purposefully used a simple model to reduce the number of free parameters and provide a conservative value of *n*. While this allows us to rule out an occupancy model and estimate the magnitude of the kinetic proofreading effect, it is not meant to reflect the specific biochemical steps in the pathway. Accordingly, our model differs from the data in some important ways. Most significantly,

the model predicts signaling to be more sensitive to CAR occupancy than we observe. Even at the lowest CAR occupancies we measured, the single cell data (*Figure 1D,E* and *Figure 3—figure supplement 1A*, top left panel) show no obvious reduction in DAG signaling, as our model predicts (*Figure 4A*, middle panel). Interestingly, more detailed models of kinetic proofreading that incorporate feedback loops can reduce the pathway's sensitivity to receptor occupancy (*Lalanne and François, 2013*) and might improve our model's fit. The system we developed here to specifically change ligand binding half-life should provide a surgical tool to help experimentally investigate these models and other mechanisms that might enhance discriminating ligands by their binding kinetics.

## Discussion

How much of ligand discrimination do our results account for? As T cells can respond to between 1–10 foreign-pMHCs (*Irvine et al., 2002*; *Huang et al., 2013*) in the context of ~$10^5$–$10^6$ self-pMHCs (*Unternaehrer et al., 2007*; *Cohen et al., 2003*; *Bhardwaj et al., 1993*), foreign-pMHCs must signal at least $10^4$–$10^5$ fold stronger than self-pMHCs. Given that differences in binding half-lives between self-pMHCs and foreign-pMHCs are on the order of 10-fold (*Davis et al., 1998*; *Germain and Stefanová, 1999*; *Gascoigne et al., 2001*; *Huang et al., 2010*), our observed degree proofreading can account for much, but not all, of ligand discrimination. T cells likely use kinetic proofreading in combination with other mechanisms, such as co-receptor involvement (*Irvine et al., 2002*), mechanical forces (*Kim et al., 2009*; *Feng et al., 2017*; *Liu et al., 2014*), or signal amplification downstream of DAG (*Das et al., 2009*), to fully discriminate self from non-self.

What biochemical steps underlie the observed kinetic proofreading? We addressed this question by measuring one of the earliest signaling events: the recruitment of ZAP70 to the CAR. ZAP70 binds to doubly phosphorylated immunoreceptor tyrosine-based activation motifs (ITMAs) in the CAR's cytoplasmic domain. In contrast to DAG levels, ZAP70 recruitment showed no detectable evidence of kinetic proofreading (*Figure 4—figure supplement 3*). In other words, ZAP70 recruitment was solely dependent on CAR occupancy and was unaffected by the ligand binding half-life. As live cell reporters reflect the cumulative effect of all upstream signaling steps, these results argue that the observed kinetic proofreading occurs somewhere downstream of ZAP70 recruitment and upstream of DAG production.

It is important to recognize how cell adhesion can affect cell signaling, as T cells adhered to a supported lipid bilayer can display ligand-independent signaling (*Chang et al., 2016*). The close (<20 nm) apposition of the two membranes can exclude the bulky transmembrane phosphatase CD45 away from the more homogenously distributed CARs or TCRs, favoring ITAM phosphorylation and T cell signaling (*Chang et al., 2016*; *Cai et al., 2018*). We initially used an anti-β2m antibody to passively adhere our cells (*Huse et al., 2007*) to ensure any decreased fluorescence of our TIRF reporters could not be due to cell detachment, but our results are not sensitive to this adhesion system. We observe an equal degree of proofreading at the level of DAG whether or not passive adhesion was used (n = 2.7 ± 0.5 with anti-β2m versus n = 2.8 ± 0.2 without anti-β2m [*Figure 4—figure supplement 3*]; 95% CI). Additionally, when cells were adhered to the SLB only with the anti-β2m antibody and no LOV2 ligand, they displayed little ruffling (*Video 4*) and lower intensities of the DAG reporter (compare *Figure 1E* and *Figure 1—figure supplement 1C*), suggesting there was no gross activation of the cells. This could be because the antibody we used did not zipper the cell membrane close enough to the SLB to effectively excluded CD45 on its own.

As the anti-β2m antibody was not necessary, we did not use the it in the ZAP70 experiments. While we did not measure CD45 localization, our results suggest that CD45 exclusion does not contribute to measurable kinetic proofreading at the level of ZAP70 recruitment in our CAR system. However, because CD45 exclusion and T cell signaling are sensitive to the size of the extracellular domains (*Chang et al., 2016*; *Cai et al., 2018*; *Irles et al., 2003*; *James and Vale, 2012*; *Choudhuri et al., 2005*; *Chen et al., 2017*), CD45's contribution to kinetic proofreading may be different in the context of the native TCR-pMHC interaction.

Even though LOV2 was the only means of cell adhesion in the ZAP70 experiments, the cells retained a small RICM footprint under the higher, inhibitory intensities of blue light. The CAR could adhere the cells by many weak interactions with the abundant LOV2 in the low-affinity conformation (*Wang et al., 2016*) or with scarce LOV2 in the high-affinity conformation. Because saturating amounts of blue light were not necessary to maximally suppress cell signaling, a small fraction of

LOV2 was likely always in the high-affinity conformation, albeit with a short half-life, even for the highest intensity blue light used with cells.

Our results differ from a study with a DNA-based CAR, which found a delay between ligand binding and ZAP70 recruitment, implicating some kinetic proofreading effects upstream of ZAP70 recruitment. Mechanistically, this delay was related to the time needed for ligand-occupied CARs to form small clusters, the only place where effective ZAP70 recruitment was observed. The putative dimeric state of our CAR (*Irving and Weiss, 1991*) versus the putative monomeric state of their CAR (*Bhatia et al., 2005*) could account for these differences.

Significantly, for cells with high CAR occupancy but low ligand binding half-life, we observe ZAP70 recruitment without high DAG levels. These results nicely match the prediction of a two-step ZAP70 activation model. Requiring ZAP70 to bind a phospho-ITAMs and then subsequently be phosphorylated by Lck for full activity allows cells to have significant ZAP70 recruitment without strong downstream signaling (*Thill et al., 2016*). It also matches observations of thymocytes and peripheral T cells in vivo, where ZAP70 can be recruited to the CD3zeta but not phosphorylated (*van Oers et al., 1994*; *Madrenas et al., 1995*), a prerequisite for full activity (*Yan et al., 2013*; *Chakraborty and Weiss, 2014*).

Previous work has shown that the recruitment of Lck to the TCR is a key rate-limiting step in discriminating ligand binding lifetimes but left open whether it was mediated through Lck's phosphorylation of ITAMs or ZAP70 (*Stepanek et al., 2014*). As we observe no kinetic proofreading in ZAP70 recruitment, our results suggest Lck's proofreading effects are mediated through ZAP70 phosphorylation. However, TCR ITAM phosphorylation is sensitive to the strength of the pMHC (*Sloan-Lancaster et al., 1994*; *Madrenas et al., 1997*; *Madrenas et al., 1995*). It could be that ZAP70 recruitment doesn't parallel ITAM phosphorylation(*Sloan-Lancaster et al., 1994*), as both tyrosines within a single ITAM need to be phosphorylated to effectively recruit ZAP70 or that parameters other than binding lifetime affect ITAM phosphorylation. Future experiments that titrate ligand binding half-life with our system and measure both ITAM phosphorylation, ZAP70 recruitment and ZAP70 phosphorylation would help parse the mechanism in finer detail.

ZAP70 phosphorylation is attractive as a proofreading step because it occurs physically close to the receptor, allowing it to reset quickly upon ligand dissociation. This is a central requirement of kinetic proofreading. Another candidate biochemical step could be the phosphorylation of Y132 on LAT because of its involvement in recruiting PLCγ (*Paz et al., 2001*; *Lin and Weiss, 2001*; *Zhang et al., 2000*). Lck was recently shown to act as a bridge between ZAP70 and LAT (*Lo et al., 2018*), suggesting that at least some fraction of LAT molecules could be tethered to phosphorylated ITAMs. Future experiments should focus on how mutations that change the kinetics of LAT Y132 or ZAP70 Y315/Y319 phosphorylation affect ligand discrimination.

In summary, we addressed a fundamental question in immunology: How do T cells discriminate ligands? We overcame previous technical limitations by developing a new optogenetic approach that directly tunes ligand binding half-life while keeping constant all other parameters that could affect CAR signaling. We found direct evidence that ligand binding half-life strongly controls downstream DAG levels but not upstream signaling events, like ZAP70 recruitment.

Additionally, our experiments with a CAR revealed a similar degree of kinetic proofreading as complementary experiments that measured calcium release downstream of optogenetic stimulation of the TCR (see *Yousefi et al., 2019*). As both DAG levels and calcium release are similarly situated in the signaling pathway (both being immediate consequences of PLCγ activity), it suggests that the observed kinetic proofreading is a consequence of shared signaling elements between these two systems, either at the level of the receptors and/or the proximal signaling pathway. Considering our ZAP70 results, the most parsimonious explanation is that ligand half-life discrimination is not completed entirely by the receptor (CAR or TCR) and that kinetic proofreading can emerge from steps in the proximal signaling pathway.

To date, optogenetics has been a powerful approach because it tells us how the timing, amount, and localization of an ensemble of signaling molecules modulate cell signaling (*Tischer and Weiner, 2014*). With the direct control over the persistence of individual protein-protein interactions presented here, optogenetics can advance to explore how the lifetime of individual molecular interactions regulates cell signaling. This could be a powerful angle for investigating other pathways where the individual molecular kinetics rather than the ensemble average are thought to gate downstream

signaling (*Huang et al., 2016*; *MacQueen et al., 2005*; *Freed et al., 2017*; *Huang et al., 2019*; *Case et al., 2019*).

# Materials and methods

## Key resources table

| Reagent type (species) or resource | Designation | Source or reference | Identifiers | Additional information |
|---|---|---|---|---|
| Recombinant DNA reagent | C1-Halo | PMID: 17629516 | | Dr. Mark M Davis (Stanford University) |
| Recombinant DNA reagent | ZAP70-mCherry | PMID: 23840928 | | Dr. Jay Groves (UC Berkeley) |
| Recombinant DNA reagent | LOV2 (V529N) | PMID: 27427858, PMID: 18604202 | | Dr. Klaus Hahn (UNC Chapel Hill) |
| Recombinant DNA reagent | Zdk1 (purified) | PMID: 27427858 | | Dr. Klaus Hahn (UNC Chapel Hill) |
| Recombinant DNA reagent | Zdk-CAR | This paper and PMID: 1705867 | | Zdk1 was fused to the N-terminus of an existing CD8 CAR, provided by Dr. Art Weiss (UCSF) |
| Cell line (*H. sapiens*, male) | Jurkat | PMID: 6327821 | RRID: CVCL_0367 | Dr. Art Weiss (UCSF) |
| Cell line (*H. sapiens*, male) | Jurkat expressing Zdk-CAR and C1-Halo reporter | This paper. | | A clonal Jurkat line expressing the Zdk-CAR and C1-Halo reporter made via lentiviral transduction. |
| Cell line (*H. sapiens*, male) | Jurkat expressing Zdk-CAR and ZAP70-mCherry reporter | This paper. | | A clonal Jurkat line expressing the Zdk-CAR and ZAP70-mCherry reporter made via lentiviral transduction. |
| Antibody | Mouse anti-human B2 microglobulin | BioLegend | Cat. #: 316308 RRID: AB_493689 | Cells labeled at 0.5 ug/ml in growth media. |
| Chemical compound, drug | PP2 | abcam | Cat. #: ab120308 | Used at 10 uM |
| Chemical compound, drug | Halo dye (JF549) | PMID: 25599551 | | Dr. Luke Lavis (Janelia Research Campus) |
| Chemical compound, drug | Alexa Fluor 488 C5 Maleimide | ThermoFisher Scientific | Cat. #: A10254 | |
| Chemical compound, drug | Sulfo-Cyanine3 maleimide | Lumiprobe | Cat. #: 11380 | |
| Chemical compound, drug | POPC | Avanti Polar Lipids | Cat. #: 850457C | |
| Chemical compound, drug | PEG-PE | Avanti Polar Lipids | Cat. #: 880230C | |
| Chemical compound, drug | biotinyl CAP PE | Avanti Polar Lipids | Cat. #: 870277X | |

## Cloning

Standard molecular biology protocols were used for all cloning. In general, individual DNA segments were amplified by PCR and assembled using the isothermal Gibson assembly method. A plasmid encoding the C1 domains of PKCθ were a kind gift from Mark M Davis (*Huse et al., 2007*). The Zdk-CAR was based on a CD8-CAR (*Irving and Weiss, 1991*), the plasmid for which was a kind gift from Art Weiss. The V529N mutation (*Yao et al., 2008*) in LOV2 biases it towards the 'open' conformation which does not bind Zdk. This mutation facilitated the quick release of LOV2 from the Zdk-CAR. See *Table 1* for a list of plasmids used in this study.

**Table 1.** Plasmids use in this study.

The plasmid name, the expressed protein, and a brief description of the construct are given. The entire coding regions of all constructs were verified by Sanger Sequencing. Plasmids and detailed maps are available from Addgene.

| Plasmid | Expressed protein | Backbone | Description |
|---|---|---|---|
| pDT326 | DAG reporter | pHR (*James and Vale, 2012*) | C1 domains mPKCθ (aa 45–166)-HaloTag |
| pDT481 | Zdk1 | pETM11-SUMO3 | 10xHis-TEV-SUMO3-KCK-SpyCatcher-GS linker-Zdk1 |
| pDT523 | ZAP70 reporter | pHR | hZAP70-mCherry |
| pDT537 | Zdk-CAR | pHR (*James and Vale, 2012*) | IgK ss-HA tag-Zdk1-GS linker-hCD8α (aa 22–208)-mCD3ζ (aa 52–164)-tagBFP |
| pDT552 | LOV2 | pETM11-SUMO3 | 10xHis-TEV-AviTag-KCK-LOV2 V529N |

DOI: https://doi.org/10.7554/eLife.42498.026

## Cell culture

Jurkat cells were grown in RPMI 1640 (Corning Cellgro, #10–041-CV) supplemented with 10% fetal bovine serum (Gibco, #16140–071) and glutamine (Gibco, #35050–061). Jurkats were maintained at densities between 0.1 and $1.0 \times 10^6$ cells per ml. 293 T cells were grown in DMEM (Gibco, #11995–065) with 10% fetal bovine serum. All cell lines were grown in humidified incubators at 37°C with 5% $CO_2$.

## Cell line construction

1 ml of wt Jurkat cells at $0.5 \times 10^6$ cells/ml were combined with 0.5 ml each lentiviral supernatant for the Zdk1-CAR and the appropriate reporter construct. Cells recovered overnight in the incubator, 8 ml of media was added the following day and cells were grown to desired density. Cells were labeled with the Halo dye JF549 (*Grimm et al., 2015*) and single cells clones were sorted by FACS (FACSAria II, BD) into 96-well plates (Corning Costar #3595) containing a 50/50 mixture of 0.22 um filtered conditioned media and fresh media. Clones were grown up for approximately three weeks and tested for blue-light dependent signaling on LOV2 functionalized SLBs on the microscope. Responsive clones with sufficient expression of both components for microscopy were saved and frozen down.

Wild-type Jurkat cells in this study were obtained from the laboratory of Dr. Art Weiss and were regularly tested for mycoplasma and found to be negative.

## Lentiviral production

Lentivirus was produced in 293 T cells using a second-generation lentiviral system (*James and Vale, 2012*). Cells grown to 40–60% confluency in 6-well plates were transfected with 0.5 ug each of pHR (containing the transgene of interest), pMD2.G (encoding essential packaging genes) and p8.91 (encoding VSV-G gene to pseudotype virus) using 6 ul of Trans-IT (Mirus, #MIR 2705) per manufacturer's instructions. (Plasmids kind gift from Ron Vale.) After 48 hr, the supernatant was filtered through a 0.22 um filter and used immediately or frozen at −80°C until use.

## Cell preparation for imaging

For each imaging well, approximately $1 \times 10^6$ Jurkat cells well labeled with the Halo dye JF549 (*Grimm et al., 2015*) (10 nM, a kind gift from the Lavis lab) for at least 15 min at 37°C. During the last 5 min, a biotinylated anti-B2 microglobulin antibody (BioLegened, #316308) was added (0.5 ug/ml final). Cells were washed twice into dPBS-BB (at 400 RCF, 4 min) and resuspended in 30 ul dPBS-BB and immediately added to a functionalized SLB.

## LOV2 purification

200 ml of LB Kan (30 ug/ml) was inoculated with E. coli BL21(DE3) transformed with a bicistronic plasmid expressing LOV2 and BirA and grown at 37°C overnight. 100 ml of the overnight culture was diluted into 1 L of TB Kan in a 2.8 L baffled flask and was shaken at 180 rpm at 37°C. At an OD600 of 0.6–0.8, the temperature was reduced to 18°C, and after 30 min media was supplemented with IPTG (250 uM), biotin (50 uM) and flavin mononucleotide (1 mM; BIO-RAD, #161–0501). After

growing overnight, all future steps were carried out on ice or at 4°C. Cells were pelleted and resuspended in 2 ml IMAC binding buffer for every 1 g wet cell pellet. Cells were homogenized in a dounce homogenizer and one Roche cOmplete Mini, EDTA-free protease inhibitor tablet was added. Phenylmethylsulfonyl fluoride was added (1 mM final) and cells were lysed on an EmulsiFlex-C3 (AVESTIN, Inc.). Lysate was spun in an ultracentrifuge (Beckman Coulter Optima L-90k ultracentrifuge) in a Ti-45 rotor at 40 k rpm for one hour. Using a peristaltic pump, the supernatant was recirculated over a 5 ml HiTrap Chelating column (GE Healthcare, #17040801) charged with $Co^{2+}$. After binding, the column was transferred to an AKTA (GE Healthcare, AKTA pure 25) and washed with IMAC binding buffer until effluent was 60mAu (A280) or less. Protein was eluted with a 10 column volume gradient (IMAC binding buffer to IMAC elution buffer) and collected in 1.5 ml fractions. Visibly yellow fractions contained LOV2 and were pooled and exchanged into IMAC binding buffer using a HiPrep 26/10 desalting column (GE Healthcare, #45-000-266). Purified 6xHis-TEV protease was added 1:10 w/w TEV:LOV2 and incubated overnight covered in tin foil.

The digested LOV2 mixture was recirculated over another HiTrap Chelating column with a peristaltic pump. The flow through was exchanged into HBS with a desalting column and concentrated to approximately 4 mg/ml with a Vivaspin protein concentrator (GE Healthcare, #28-9323-60). LOV2 was preferentially labeled on the N-terminal KCK tag (*Hansen and Mullins, 2010*) by adding Alexa Fluor 488 C5 Maleimide (ThermoFisher Scientific, #A10254) freshly dissolved in anhydrous DMSO (10 mM; ThermoFisher Scientific, #D12345) at a final molar ratio of 2:1 dye:LOV2. Reaction proceeded on ice for 30 s and was quenched with DTT (10 mM final). Excess dye was removed by running the mixture over a Superdex 200 Increase 10/300 GL column (GE Healthcare) equilibrated with HBS. LOV2 was concentrated to approximately 1.5 mg/ml and glycerol added to a 10% final volume. 3 ul aliquots were snap frozen in liquid nitrogen and stored at −80°C.

## Zdk1 purification
Dye labeled Zdk1 was purified in an identical manner to LOV2, excepting the following changes.

1. Bacterial growth media was not supplemented with biotin or flavin mononucleotide.
2. The SUMO protease 6xHis-SenP2 is added to the IMAC eluted protein at 1:1000 w/w instead of TEV protease.
3. Protein was labeled with Sulfo-Cyanine3 maleimide (Lumiprobe, #11380) instead of Alexa Fluor 488 C5 Maleimide.

At various stages, fractions were collected and analyzed by SDS-PAGE (*Figure 2—figure supplement 1*). Final protein preparations were free of obvious contaminating proteins.

## Media and buffers
IMAC binding buffer: 50 mM KH2PO4, 400 mM NaCl, 0.5 mM βME, pH 7.5.
IMAC elution buffer: 50 mM KH2PO4, 400 mM NaCl, 500 mM imidazole, 0.5 mM βME, pH 7.5.
HBS: 20 mM HEPES, 100 mM KCl, 0.5 mM TCEP, pH 7.5.
Terrific Broth (TB) – *Nutrient Base:* 12 g tryptone, 24 g yeast extract, 4 ml glycerol, 900 ml ddH$_2$O.
*10x TB Salts:* 170 mM KH$_2$PO$_4$, 720 mM K$_2$HPO$_4$. Autoclave the nutrient base and 10x TB salts separately. Then add 100 ml of 10x TB salts to 900 ml of the nutrient base to make 1L of TB.

## Preparation of small unilamellar vesicles (SUVs)
SUV preparation and glassware cleaning protocols were modified from those previously described (*Taylor et al., 2017*). A precleaned 4 ml glass vial was washed 2x with chloroform (Electron Microscopy Sciences, #12550) and then approximately 500 ul of chloroform was added. Using Hamilton syringes (Hamilton Company, Gastight 1700 series, #80265 and #81165), 4 umoles of lipids were added in the following molar ratio: 97.5% POPC, 0.5% PEG-PE and 2% biotinyl CAP PE (Avanti Polar Lipids, #850457C, #880230C, #870277X, respectively). Chloroform was removed by slowly rotating the vial at an angle while slowly flowing nitrogen gas (Airgas, #NI 250). The vial was loosely covered with a cap and placed in a vacuum desiccator overnight. Lipids were rehydrated with 1.5 ml of 0.22 um filtered dPBS (ThermoFisher Scientific, #14190144) and gently vortexed for 10 min. The liquid was transferred to a 1.5 ml Eppendorf tube and closed under nitrogen gas. Lipids were cycled between freezing in liquid nitrogen and thawing in a 42°C water bath 20 times. The mixture was

spun at 60 k RCF for 40 min at 4C (Beckman Coulter Optima MAX-TL Ultracentrifuge). The supernatant was removed and stored in liquid nitrogen until ready for use.

## Glassware cleaning

Glass vials (ThermoFisher Scientific, #B7800-2) and pasteur pipettes (ThermoFisher Scientific, #13-678-20A) were added to a 1L beaker, covered with 3M NaOH and bath sonicated (Branson 1800) for 30 min. The NaOH was decanted and glassware was washed five times with ddH$_2$O. Glassware was covered with a 5% (v/v) Hellmanex III (Sigma-Aldrich, #Z805939-1EA) solution and incubated overnight. The Hellmanex solution was decanted and the glassware was extensively washed with ddH$_2$O to remove all traces of the detergent. Excess water was removed by blow drying with nitrogen gas. Glassware was then dried at 80°C and stored protected from dust. Vial caps were sonicated in ddH$_2$O for 30 min, dried and stored with the glassware.

## RCA cleaning of microscopy coverslips

Glass coverslips (Ibidi, #10812) were added to a glass Coplin jar (Sigma-Aldrich, #BR472800) and successively bath sonicated for 10 min each in acetone (Sigma-Aldrich, #534064–4L), 190 proof ethanol (Koptec, #V1101), and ddH$_2$O. Coverslips were washed five times in ddH$_2$O before and after the ddH$_2$O bath sonication to remove excess organic solvents. 3.75 g KOH was dissolved in 45 ml ddH2O and added to the coverslips, followed by 15 ml 30% hydrogen peroxide (Fisher Scientific, #H325-500). The Coplin jar was placed into a 70–80°C water bath and allowed to react for 10 min. The base solution was decanted and coverslips were washed five times in ddH$_2$O. The following were added to the coverslips in order: 38 ml ddH2O, 9.5 ml 37% HCl (Acros, #42379–5000) and 12.6 ml 30% hydrogen peroxide. Coverslips were incubated for 10 min in the water bath. The acid solution was decanted, coverslips were washed five times in ddH$_2$O and stored in ddH$_2$O for up to one week.

## Functionalization of SLBs and cell preparation

An RCA cleaned glass coverslip was removed from ddH2O and immediately blown dry with compressed nitrogen. A six-well Ibidi sticky chamber (Ibidi, #80608) was firmly pressed on top. Edges were sealed with clear nail polish and let set for five minutes. A 30 ul SUV aliquot was diluted with 800 ul 0.22 um filtered dPBS, added to each well and then incubated at 37°C for one hour. To functionalize a well, excess lipids were flushed out with 500 ul 0.22 um filtered dPBS. Streptavidin (Rockland, #S000-01) diluted in dPBS-BB (2 ug/ml final) was flushed into the well and incubated at room temperature for 5 min. The well was flushed with 500 ul dPBS-BB. LOV2 diluted in dPBS-BB (typically between 20–200 nM) was flushed into the well and incubated in the dark at room temperature for 5 min. The well was then flushed with 500 ul dPBS-BB and incubated with cells previously labeled with the halo dye and biotinylated anti-β2 microglobulin antibody (BioLegend, #316308) and washed twice into dPBS-BB. Cells adhered to the SLB in the dark for 5 min. The well was flushed with 500 ul of imaging media and transferred to the microscope for imaging.

## Buffers for SLB functionalization and imaging

dPBS-BB: dPBS (ThermoFisher Scientific, #14190144) with 1 mg/ml beta-casein and 0.5 mM βME. Filtered with a 0.22 um filter prior to use.

Imaging media: RPMI-1640 without phenol red (Gibco, #11835–030) supplemented with 1% fetal bovine serum, 1:100 dilution of Glutamax (Gibco, #35050–061), 10 mM HEPES (pH 7.4), 1 mg/ml β-casein, 0.5 mM beta mercaptoethanol, 50 ug/ml ascorbic acid and 1:100 dilution of ProLong Live Antifade Reagent (ThermoFisher Scientific, #P36975). Solution was passed through a 0.22 um filter and incubated at room temperature for at least 90 min to allow the antifade reagent to reduce oxygen levels.

## Microscopy

Imaging was performed on an Eclipse T*i* inverted microscope (Nikon) with two tiers of dichroic turrets to allow simultaneous fluorescence imaging and optogenetic stimulation. The microscope was also equipped with a motorized laser TIRF illumination unit, a Borealis beam-conditioning unit (Andor Technology), a CSU-W1 Yokogawa spinning disk (Andor Technology), a 60x Apochromat

TIRF 1.49 NA objective (Nikon), an iXon Ultra EMCCD camera, an Evolve 512 EMCCD camera (Photometrics) and a laser merge module (LMM5; Spectral Applied Research) equipped with 405-, 440-, 488-, 514-, and 561 nm laser lines. For RICM, light from a Xenon arc lamp (Lambda LS, Sutter Instrument) source was passed through a 572/35 nm excitation filter (Chroma, #ET572/35x) filter and then a 50/50 beam splitter (Chroma, #21000). Microscope and associated hardware was controlled with MicroManager (*Edelstein et al., 2014*) in combination with custom built Arduino controllers (Advanced Research Consulting Corporation). Blue light for optogenetic stimulation was from a 470 nm LED (Lightspeed Technologies Inc., #HPLS-36), independently controlled with MATLAB. The maximum power density cells were exposed to was 16 uW/cm$^2$. For most timepoints, only RICM and TIRF561 images were collected. During and in between these timepoints, a TIRF488 dichroic mirror remained permanently in the top dichroic turret, ensuring the blue-light illumination of the cells was never interrupted. The top TIRF488 dichroic passed the longer wavelengths used for RICM and TIRF561. Only when LOV2 localization was imaged with TIRF488 at the end of a three-minute stimulation was the top dichroic removed to allow the shorter fluorescence excitation light to pass.

## Image processing

ImageJ (National Institutes of Health) was used for all image manipulation and measurements, while MATLAB was used to analyze the measurements. Prior to analysis, all imaging channels underwent drift correction (K. Li, 'The image stabilizer plugin for ImageJ,' http://www.cs.cmu.edu/~kangli/code/Image_Stabilizer.html, February, 2008) and flat field correction. After each day of imaging, TIRF488, TIRF561 and RICM images were taken on slides with concentrated solutions of fluorescein, Rose Bengal or dPBS, respectively. Camera offset was determined by taking images with the shutter closed. To flat field correct, the camera offset was subtracted from both the experimental and dye images. The experimental image was then divided by the dye slide image to yield the final flat field corrected image used in analysis.

## Time course overview

Typically, cells were held in strong blue light for five minutes and then exposed to five minute blocks of stimulation. Each block was composed of an initial two minute hold in strong blue light followed by a three minute stimulation at a fixed intensity of intermediate blue light (*Figure 2—figure supplement 2*). DAG levels reported are the average of the final six timepoints. Since each timepoint is 10 s apart, these timepoints are acquired during the last minute of the three minute stimulation. CAR occupancy was measured from a TIRF488 image taken at the end of the three minute stimulation, after the last DAG measurement made in TIRF561. As fluorescence excitation light from TIRF488 potently stimulates LOV2, the TIRF488 channel could only be imaged once at the very end of a three minute stimulation. These five minute blocks were repeated over the course of an hour to stimulate cells with a variety of blue light intensities.

Cells were illuminated with blue light intensities that were 'out of order' to help guard against any time-dependent 'position effects' or changes of state of the cells. The two minute pulse of strong blue light before each three minute stimulation was intended to help keep each stimulation independent and guard against hysteresis by beginning each three minute stimulation from the same low-signaling state.

## DAG measurements

To calculate DAG levels at steady state, the mean TIRF561 pixel intensity within the cell mask was averaged over the last six frames (equivalent to the last 50 s) of a three-minute hold in blue light. To account for background fluorescence from the DAG reporter in the absence of signaling, cells were treated with saturating amounts of PP2 (10 uM final; abcam, #ab120308) after every time course. The resulting mean TIRF561 pixel intensity was subtracted from mean TIRF561 pixel intensity in response to blue light (see *Figure 2—figure supplement 2*). We sometimes observed minor drift in TIRF561 values over time as cell slowly contracted. This drift was stimulation independent (*Figure 1—figure supplement 1C*) and could obscure relatively small increases in DAG signaling. To correct for this drift, every three-minute stimulation of blue light was preceded by a standard two minute 'reset' pulse of fixed, higher intensity blue light. The magnitude of the drift was measured as the difference between mean TIRF561 pixel intensity of a reset pulse and the average mean TIRF561 pixel intensity

of the last three reset pulses (when the drift had stabilized). The magnitude of the drift was subtracted from the PP2 corrected mean TIRF561 pixel intensities to yield the raw DAG level.

DAG values were either normalized relative to saturation or to treatment with the Src family kinase inhibitor, PP2. See *Figure 4—figure supplement 2* for an explanation of DAG normalization schemes.

## CAR occupancy measurements

RICM images were thresholded to create a mask of the cell footprint. A second local background mask was made by constructing a thin ribbon along and slightly expanded from the perimeter of the cell mask (*Figure 2—figure supplement 3A*). Once steady state is reached, free LOV2 should be homogenously distributed on the SLB. The mean TIRF488 pixel intensity the cell footprint is the sum of free LOV2 and LOV2 bound to the CAR, while the mean TIRF488 pixel intensity in the background mask reflects free LOV2. Therefore, CAR occupancy was calculated as the mean TIRF488 pixel intensity in the cell mask minus the mean TIRF488 pixel intensity in the background mask.

We noticed fluctuations in the 488 nm laser power on the timescale of seconds. To minimize the errors in quantitation from these fluctuations, the 488 nm laser power was tracked by measuring relative changes the integrated whole field TIRF488 fluorescence (*Figure 2—figure supplement 3B*). On a large scale, the TIRF488 fluorescence should remain constant (as the LOV2-Alexa488 can only diffuse laterally on the SLB), only being affected by changes in 488 nm laser power and photobleaching. Integrated whole field TIRF488 fluorescence intensities were first normalized to their mean over the time course. Measured CAR occupancies were then divided by the corresponding normalized whole field TIRF488 fluorescence. This procedure significantly reduced the variability of measured CAR occupancies.

## ZAP70 measurements

Imaging of the ZAP70 reporter was done similarly to the DAG reporter, except the anti-β2 microglobulin antibody was omitted. We observed greatly reduced dynamic range of the ZAP70 reporter when imaged with the anti-β2 microglobulin antibody due to high basal recruitment. However, in the absence of the antibody, the cell TIRF footprint was small during the PP2 wash-in experiments (the passive adhesion of the anti-β2 microglobulin antibody helps maintain larger footprints when cells are not stimulated), which led to high variability in the mean pixel intensity of basal ZAP70 recruitment. Therefore, ZAP70 recruitment is reported simply as the mean intensity of pixels within the cell footprint, as determined from the RICM images. The corresponding DAG data sets were acquired and analyzed in the same way as the ZAP70 data to ensure that our DAG kinetic proofreading results were not altered by the absence of the anti-β2 microglobulin antibody.

## Criteria for including or excluding cells in analysis

All cells visible within the TIRF field we analyzed, only excluding cells that were dead or dying (judge morphologically) or detached from SLB during the acquisition.

## Biological and technical replicates

A biological replicate consisted of four to five time courses (see *Figure 2—figure supplement 2*) of stimulating cells with blue light on SLBs with different concentrations of LOV2, all on the same day (to ensure the light path did not change). Each biological replicate was conducted on different days, with new preparations of cells, SLBs and LOV2. Each time course within a biological replicate contained approximately 30 cells, whose DAG levels and receptor occupancy were individually measured. As the microscopy experiments could not be done in parallel and each biological replicate took an entire day, there are no technical replicates.

## LOV2 binding half-life measurements

Supported lipid bilayers were functionalized with saturating amounts of LOV2 and imaging media (see SLB functionalization and cell prep) containing 250 nM purified Zdk-Cy3 was added. To measure Zdk unbinding kinetics, wells were quickly flushed with 200 ul of imaging media without Zdk and the blue LED was immediately turned on. TIRF561 images were acquired every 200 ms. The apparent dissociation rate constant ($k_{obs}$) was measured by fitting a single exponential decay to the data after

the blue LED turned on. All measurements were performed at 37°C. Zdk was modeled as being able to unbind directly from the high-affinity state of LOV2 or quickly unbind from the low-affinity state of LOV2 after a light-dependent conformational change in LOV2.

LOV2 + Zdk LOV2 + Zdk

$\uparrow k_3$ $\uparrow k_2$

LOV2·Zdk $\xrightarrow{k_1 v}$ LOV2·Zdk
(high affinity) (low affinity)

**Scheme 1.** Zdk dissociation from LOV2.
DOI: https://doi.org/10.7554/eLife.42498.027

In such a model,

$$k_{obs} = \frac{k_1 v \cdot k_2}{k_1 v + k_2} + k_3 \tag{1}$$

where $v$ is the intensity of blue light. $k_{obs}$ was measured at various blue light intensities and subsequently fit to *Equation 1* by least squares. Blue light intensity was experimentally measured as a function of LED voltage to account for non-linear behavior at low LED voltages. The effective LOV2 binding half-life that cells experienced were calculated from this model ($\tau = \ln(2)/k_{obs}$).

## Kinetic proofreading model fitting

Single cell data of CAR occupancies, binding half-lives and DAG levels were fit to a basic model of kinetic proofreading, consisting of $n$ number of strong proofreading steps (i.e. – the forward reaction is sufficiently slow that the probability of completing a step is proportional to the ligand binding half-life) that occur upstream of a saturable DAG response.

In kinetic proofreading (*McKeithan, 1995*), the receptor transitions through several intermediate states ($C_i$) *en route* to its active form ($C_n$). The steps between each of these states are irreversible (i.e. – they consume energy) and only occur while the ligand (L) is bound to the receptor (P). Upon ligand dissociation, the receptor resets to its initial state. Intermediate states are in kinetic competition to either progress to the next state or reset upon ligand dissociation.

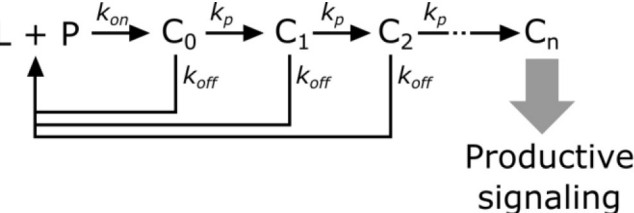

**Scheme 2.** Kinetic proofreading steps.
DOI: https://doi.org/10.7554/eLife.42498.028

The probability of a receptor progressing from one state to the next is

$$\alpha = \left(\frac{k_p}{k_{off} + k_p}\right) \tag{2}$$

Converting from rate constants to half-lives ($k = \ln(2)/\tau$) yields

$$\alpha = \left(\frac{\tau_{off}}{\tau_{off} + \tau_p}\right) \tag{3}$$

The speed of transitioning to the next step ($\tau_p$) determines how strongly each step discriminates the ligand binding half-life. When the forward reaction is fast ($\tau_p \ll \tau_{off}$), there is weak half-life discrimination, as all values of $\tau_{off}$ give a high probability of progressing to the next step. When the forward reaction is slow ($\tau_p \gg \tau_{off}$), there is strong half-life discrimination and $\alpha \propto \tau_{off}$.

The fraction of ligand-bound receptor that make it through $n$ proofreading steps is,

$$f_R = \alpha^n \tag{4}$$

Without knowing the value of $\tau_p$, it is near impossible to constrain the value of $n$, as the same fraction of receptors can be active as the result of many weak proofreading steps or few strong proofreading steps. However, assuming that all steps strongly proofread puts a minimum bound on the value of $n$. This assumption could be relaxed if we could estimate $\tau_p$, though it is not necessarily the same for all steps in the kinetic proofreading pathway.

The amount of receptors productively signaling ($S$) is

$$S = R \cdot \alpha^n \tag{5}$$

where $R$ is the total amount of ligand-bound receptors.

Finally, we model DAG levels as being saturable and downstream of the kinetic proofreading portion of the signaling pathway.

$$DAG \propto \frac{R \cdot \alpha^n}{K + R \cdot \alpha^n} \tag{6}$$

Under the strong proofreading assumption to constrain the minimum value of $n$, this becomes

$$DAG \propto \frac{R \cdot \tau_{off}^n}{K + R \cdot \tau_{off}^n} + \beta \tag{7}$$

where $K$ is the amount of upstream signal that gives half-maximal DAG response and $\beta$ is basal signaling in the pathway.

To fit the model, data sets acquired on the same day but with different amounts of LOV2 on the SLBs were combined and fit using non-linear least-squares regression as implemented by the curve_-fit function from the SciPy library. Because CAR occupancy was measured in arbitrary units, the values are sensitive to changes in the microscope's optical path. This precluded combining data sets from different days. Because all fits had degrees of freedom well in excess of 30, the distribution of the fitted variables was treated as a normal distribution. The 95% confidence interval on $n$ was calculated as $n \pm 1.96 \cdot \sqrt{\mathrm{cov}(n)}$.

*Figure 4—figure supplement 1* shows the DAG plots and values of $n$ for all data sets and normalization conditions.

## Absolute CAR and TCR quantification

Beads with known antibody binding capacity (Quantum Simply Cellular, Bangs Lab #815) were used to calculate the number of CAR and TCR molecules on the cell surface for several cell lines. Cells were first washed in PBS and fixed for 10 min in a 1:1 mixture of PBS:BD Cytofix at room temperature. Cells were washed twice with FACS buffer and stained with the primary antibody (either OKT3 or OKT8; UCSF antibody core) for 30 min at room temperature. Cells were again washed twice with FACS buffer and stained with the secondary antibody (Goat anti-mouse Alexa488, Invitrogen A-11029). Cells were washed twice with FACS buffer and analyzed by flow cytometry (BD LSR II). An identical staining protocol was used for the Quantum Simply Cellular beads, except they were not fixed. Care was taken to ensure both the primary and secondary antibodies were saturating by using a series of 2-fold serial dilutions of one antibody while the other was held at the highest concentration. No significant decrease of fluorescence was observed for the first 2-fold dilution of either the primary or secondary antibodies, confirming the highest concentration reached saturation. The highest, saturating concentration of OKT3 was 50 ug/ml, OKT8 was 10 ug/ml and the secondary antibody was 20 ug/ml. FACS buffer: dPBS (Ca2+/Mg2 +free; ThermoFisher Scientific, #14190144) with 1% (v/v) FBS and 2 mM EDTA.

Total surface receptor numbers were calculated per manufacturer's instructions. In short, log log regression of antibody binding capacity as a function of bead MFI ($R2 = 0.9937$) was used to convert the MFI of each cell line to total number of bound primary antibodies. Background antibody binding in the no primary control was then subtracted. Because the IgG primary antibodies can bind between one and two surface receptors, a two-fold range is reported.

## Absolute concentration of LOV2 on the bilayer

To estimate the absolute molecular density of LOV2 in our bilayer, we used single molecule TIRF microscopy.

First, the LOV2 binding capacity of the bilayer at saturation was calculated. Limiting amounts of LOV2-Alexa488 were diluted into LOV2-Cy3 and added to bilayers already functionalized with saturating amounts of streptavidin until single molecules of LOV2-Alex488 were resolvable. The number of LOV2-Alexa488 molecules were counted by hand and the total LOV2 molecules on the bilayer were calculated knowing the dilution factor, original concentrations of LOV2-Alexa488 and LOV2-Cy3 and their labeling efficiencies (measured spectroscopically).

$$\rho_{LOV2} = \frac{\rho_{Alexa488} \cdot DF \cdot \left[LOV2_{Cy3}\right]}{f_{Alex488} \cdot \left[LOV2_{Alex488}\right]}$$

Where $\rho$ is the molecular density on the bilayer (molecules/um$^2$), DF is the LOV2-Alexa488 dilution factor, [LOV2Cy3] and [LOV2Alexa488] are the concentrations of the undiluted protein stocks and fAlexa488 is the fraction of the LOV2-Alexa488 protein stock labeled with Alexa488. Saturation at these LOV2 concentrations was verified by diluting the LOV2 solution 2-fold and measuring no observable decrease in the LOV2-Cy3 intensity.

Second, to estimate the LOV2 density during cell-based experiments, we calculated the fraction of saturation achieved with the various concentrations of LOV2 used to functionalize the bilayers. The product of the fraction of saturation and LOV2 density at saturation yielded the estimates of absolute LOV2 density under experimental conditions.

## Acknowledgements

We thank Dyche Mullins and Peter Bieling for help with biochemical purifications, Jay Groves, Geoff O'Donoghue, Scott Hansen and Marcus Taylor for help with the supported lipid bilayers, and Sam Lord for help with RICM. Thanks to Klaus Hahn, Orrin Stone, Torsten Wittmann, and Jeffrey van Haren for help troubleshooting the LOVTRAP system. Jared Toettcher, Elliot Dine, Hana El-Samad and Art Weiss provided critical reading of the manuscript. This work was supported by a Genentech Fellowship (DT), NIH grants GM109899 and GM118167 and the Novo Nordisk Foundation (ODW) and the Center for Cellular Construction (DBI-1548297), an NSF Science and Technology Center.

## Additional information

### Funding

| Funder | Grant reference number | Author |
|---|---|---|
| Genentech Foundation | Graduate Student Fellowship | Doug K Tischer Orion David Weiner |
| National Institutes of Health | GM109899 | Orion David Weiner |
| Novo Nordisk | | Orion David Weiner |
| National Science Foundation | DBI-1548297 | Orion David Weiner |
| National Institutes of Health | GM118167 | Orion David Weiner |

The funders had no role in study design, data collection and interpretation, or the decision to submit the work for publication.

### Author contributions

Doug K Tischer, Conceptualization, Software, Funding acquisition, Investigation, Visualization, Methodology, Writing—original draft, Writing—review and editing; Orion David Weiner, Conceptualization, Resources, Funding acquisition, Writing—original draft, Writing—review and editing

Author ORCIDs

Doug K Tischer http://orcid.org/0000-0003-3633-5542

Orion David Weiner http://orcid.org/0000-0002-1778-6543

Decision letter and Author response

Decision letter https://doi.org/10.7554/eLife.42498.031

Author response https://doi.org/10.7554/eLife.42498.032

## Additional files

### Supplementary files

• Transparent reporting form

DOI: https://doi.org/10.7554/eLife.42498.029

### Data availability

All single cell measurements generated or analysed during this study are included in the manuscript and supporting files. Source data files have been provided for Figure 4, Figure 4-figure supplement 1 and Figure 4-figure supplement 3.

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
