## [Decision Letter]

Thank you for submitting your article "Light-based tuning of ligand half-life supports kinetic proofreading model of T cell activation" for consideration by *eLife*. Your article has been reviewed by Arup Chakraborty as the Senior Editor, a Reviewing Editor, and two reviewers. The reviewers have opted to remain anonymous.

The reviewers have discussed the reviews with one another and the Reviewing Editor has drafted this decision to help you prepare a revised submission.

Summary:

In this system the Lov2 domain binds Zdk1 with a low nM Kd in the dark and a low µM Kd in with intense blue light. Intense blue light will induce rapid conversion to low affinity form and dimmer blue light will induce slower conversion (even when Zdk1 is bound) and some degree of dark recovery of those molecules equilibrate to a concentration of active Lov2 that binds Zdk1 and dissociates with a k_off_ defined by the light intensity. The Lov2 domain is attached to supported lipid bilayer (SLB) with a fluorophore and the Zdk1 domain (a small three helix bundle based on protein A repeat) is expressed as a first gen CAR. The system is cycled between the dark/dim (binding on) or light (binding off) states with the dark/dim periods leading to increased cell spreading. Even in the light state a contact area is maintained, probably by the low affinity interaction of the off Lov2 with Zdk1. As a readout the authors mostly focus on DAG sensor readout as the ZAP-70 readout for CAR itself seems to be recruited in simple proportion to engagement (even in light/off state?). The DAG reporter signal is proportional to kinetics rather that occupation. Suggesting that the step between the ITAM mediated recruitment of ZAP-70 and the LAT dependent recruitment of PLCγ (based on literature) can be involved in kinetic proofreading with a threshold around 10 seconds.

As a technical paper that develops a tool, this is an interesting paper and could be published in the Tools and Resources section. The paper is more relevant to synthetic biology and design of CAR based therapies, but limited relevance to how a TCR works or self/non-self discrimination. We do not think that the paper resolves the latter question, which the authors set out to address. A number of technical points about the method need to be addressed. Furthermore, somewhat disturbingly, the vast literature in the field is cited in a selective manner. Below, we provide guidance on both these important issues.

Essential revisions:

1) The Abstract and Introduction should be reshaped to reflect more of the background of CARs and T cell ligand discrimination and what a simplified signaling model with light based kinetic control could teach us.

2) Many aspects of the system are poorly described- at this point in the technologies they are working with they should provide:

a) Absolute quantification (molecules/cell) of the CAR on the Jurkat cells. This could be done by flow cytometry with 488 or 640 labelled Luv2 (dark) and MESF beads from Bangs labs.

b) The density of the Lov2 domain in the SLB in molecules/µm2 across the range that they are using. This could be done by line scanning FCS (dilute fluorescent version with non-fluorescent version if density is too high.) See PMID: 19254560.

c) The estimate of occupancy will be inaccurate if they don't include a control for exclusion of non-binding Luv2 mutant or similar sized protein in the bilayer with a difference fluorophore. See: PMID: 17911103.

3) It would be useful to see at least diffraction limited higher magnification images of the different signals to establish the correlation between IRM contact, CAR occupancy, the ZAP-70 and C2 domain sensor. It the CAR forming microclusters or is forming larger domains as it appears in the zoomed out views provided.

4) A number of relevant papers need to be cited that are pertinent, and the present results related to these studies. A small sampling of the missing literature are:

a) The work from the Davis lab from several years ago on using caged pMHCs that were sensitive to light, which allowed measurements that described the speed at which signals propagate downstream.

b) The role of rebinding that allows ligands with short half-lives to signal, described by work from eight years ago by Van der Merwe and co-workers (Immunity) and Huseby and co-workers.

c) The role of co-receptors, amount of active Lck and diffusion in kinetic proof reading described by Stepanek et al., (2014).

d) You also ignore the data the Germain lab, and others, have presented showing that the same ab TCR signals differently in DP thymocytes vs. mature cells [in particular, allows meaningful propagation of signals for gene expression in DP with pMHC ligands incapable of electing the same response or biochemical changes in a mature T cell], meaning that intracellular events can control signal propagation separately from occupancy, half-life or mechanosensing.

5) The model for proofreading is a bit strange. Many people have made models that are far more realistic and incorporate feedback, including several papers from the Germain lab (1995; 1997; 2005; 2002), Chakraborty and Palmer, Chakraborty and Weiss, and the more theoretical efforts by Murugan, Huse, and Leibler. These papers should at least be cited and briefly discussed.

6) The estimate of 3 proofreading steps is undoubtedly wrong. The various steps do not have the same time scales and there are feedback loops. These effects make the model's fit to obtain n very difficult to interpret. This part needs to be deleted.

7) The comments about ZAP70 not being subject to kinetic proof reading in the Introduction need to be made with great care. Two points to note in this regard: a] There are data suggesting that partial phosphorylation of ITAMs is at least correlated if not connected to antagonism. b] in vivo, ZAP that is bound to TCR is not phosphorylated. Also note that in vivo the T cells are constantly stimulated by self ligands (see recent paper by Weiss, Chakraborty and co-workers in MCB).

[Editors' note: further revisions were requested prior to acceptance, as described below.]

Thank you for resubmitting your work entitled "Light-based tuning of ligand half-life supports kinetic proofreading model of T cell signaling" for further consideration at *eLife*. Your revised article has been favorably evaluated by Arup Chakraborty (Senior Editor) and a Reviewing Editor.

The manuscript has been improved but there are some remaining issues that need to be addressed before acceptance, as outlined below:

The reviewing editor has asked that you address one point prior to a final decision. Please describe the steps in optimisation of the adhesion system and the specific requirement for the system chosen. Also, the implications of this adhesion system for kinetic proofreading – aka kinetic segregation model – should be discussed. For example, could the lack of kinetic proofreading in TCR proximal signaling be due to the presence of the adhesion system, which takes care of CD45 exclusion and makes signaling a matter of occupancy? If the light dependent kinetics were in control of CD45 segregation, would this step show a sensitivity to kinetics?

Recent relevant papers include:

Chang.et al., 2016.

Cai.et al., 2018.

The editor is not asking for additional experiments, just a transparent discussion of the results of the adhesion system, why anti-β2m, and if this is actually required for the signaling by the light inactivated CAR.

Reviewer #1:

The study continues to struggle with proving "kinetic proofreading" (KP) due to the issue that the light dependent acceleration of dissociation of the LOV2 domain from Zdk1 doesn't impact the dark reversion rate (on rate) so the affinity is decreased by blue light. This is a similar limitation faced by people working with mutation-based systems for TCR or CARs. The mass action equation makes it very hard to change the half-life while maintaining a constant affinity and occupation. They partly compensate for this by doing the experiments at different densities of the LOV2 domain to have matched receptor occupancy at all light intensities. These densities are now specified, but there is a surprise there. This is partly successful as they do see a better correlation of DAG to kinetics rather than occupation, but it’s expected that this system fires fully at low occupancy, so this lack of linear relationship to occupation is expected. Thus, the proof of KP is not advanced that much in practice, but the potential is there. Additional innovations seem necessary to fully exploit it.

Essential Revisions:

The authors have added a synthetic adhesion system, which was only mentioned in the methods and some legends. They use a biotinylated mAb to β2m, which should pull the cells very close to the substrate, perhaps less than 10 nm. I assume this will exclude CD45 at least partially if not quite robustly (the authors should check this) so all they need to do to trigger the CAR is pull them into these close contacts. This changes everything as they are not making the LOV2-Zdk1 interaction achieve KP through "kinetic-segregation". This is thought by many to be a critical process that needs to be mediated by the ITAM bearing receptor to initiate signalling in many contexts. So, this adhesion system and its implications need to be described in the experimental design and schematics in figures as its likely critical to interpretation. Can the system work at all without this synthetic adhesion system? This may explain why they see no kinetic proofreading at the level of ZAP-70- as its recruitment is taken care of by the adhesion system and then bringing the CAR into the close contacts.

The other surprise is that the binding of the LOV2 domain displays significant positive cooperativity in binding to the SLB, suggesting that it oligomerises. How does this impact the measurements and interpretation?

[Editors' note: further revisions were requested prior to acceptance, as described below.]

Thank you for submitting your article "Light-based tuning of ligand half-life supports kinetic proofreading model of T cell signaling" for consideration by *eLife*. Your revised article and response to our request has been reviewed by a Reviewing Editor and Arup Chakraborty as the Senior Editor.

Unfortunately, your response has not addressed the only issue that needs to be resolved. When the question about adhesion first came up, we asked about low affinity interaction in the off state that might support residual adhesion. You noted that this was not a relevant question because the adhesion system was added to make the system independent of the CAR interactions with the ligand to keep the cells in place over many cycles. Now you say the adhesion system is not needed and in fact you did not use it for the studies looking at ZAP-70's role in kinetic proofreading. Given this, unfortunately, we need to ask again if this indicates that there is a low affinity interaction in the 100% "off" state that could mediate some residual adhesion, but without signaling. If the cells remain attached in the presence of the most intense blue light that would completely turn off the interaction, then it seems there would be support for the presence of such an interaction. Can you acknowledge this, or do you have another explanation for the system working with or without the parallel adhesion system?

---

## [Author Response]

Summary:In this system the Lov2 domain binds Zdk1 with a low nM Kd in the dark and a low µM Kd in with intense blue light. Intense blue light will induce rapid conversion to low affinity form and dimmer blue light will induce slower conversion (even when Zdk1 is bound) and some degree of dark recovery of those molecules equilibrate to a concentration of active Lov2 that binds Zdk1 and dissociates with a k_off_ defined by the light intensity. The Lov2 domain is attached to supported lipid bilayer (SLB) with a fluorophore and the Zdk1 domain (a small three helix bundle based on protein A repeat) is expressed as a first gen CAR. The system is cycled between the dark/dim (binding on) or light (binding off) states with the dark/dim periods leading to increased cell spreading. Even in the light state a contact area is maintained, probably by the low affinity interaction of the off Lov2 with Zdk1. As a readout the authors mostly focus on DAG sensor readout as the ZAP-70 readout for CAR itself seems to be recruited in simple proportion to engagement (even in light/off state?). The DAG reporter signal is proportional to kinetics rather that occupation. Suggesting that the step between the ITAM mediated recruitment of ZAP-70 and the LAT dependent recruitment of PLCγ (based on literature) can be involved in kinetic proofreading with a threshold around 10 seconds.As a technical paper that develops a tool, this is an interesting paper and could be published in the Tools and Resources section. The paper is more relevant to synthetic biology and design of CAR based therapies, but limited relevance to how a TCR works or self/non-self discrimination. We do not think that the paper resolves the latter question, which the authors set out to address. A number of technical points about the method need to be addressed. Furthermore, somewhat disturbingly, the vast literature in the field is cited in a selective manner. Below, we provide guidance on both these important issues.

We would like to thank you and the other reviewers for providing thoughtful, constructive feedback. The experiments and literature references you shared helped to make our paper stronger and clearer. We have outlined the most significant changes to the paper below and provided a detailed point-by-point response separately.

1) The reviewers wanted us to focus more explicitly on our use of a CAR, both in the setup and interpretation of our results. During revision, we made modifications throughout the manuscript to emphasize our use of a CAR more directly. We included comparisons to historic insights from chimeric antigen receptors and the increased medical relevance of understanding their signaling mechanisms. In interpreting our results, we explicitly delineate what conclusions apply to CAR signaling and which ones could be more general features of T cell signaling.

2) Experimentally, the reviewers wanted a more detailed description of the performance of our new system, especially in terms of absolute numbers of molecules. Happily, we were able to address all of their points. Absolute quantification of the levels of cell-surface CARs (62-238 x10^3^ molecules/cell) and LOV2 densities (between 40-1200 molecules/um^2^) showed that our system performs in similar molecular ranges as previously published CAR-based approaches. High magnification images showed that both of our reporters localized as expected: ZAP70 co-localized with areas of CAR occupancy while DAG was found roughly diffusely throughout the cell footprint. To address concerns about inaccurate receptor occupancies as a result of unbound LOV2 molecules being excluded from the cell footprint, we reanalyzed our data under a worst-case scenario: that all unbound LOV2 molecules are excluded from the footprint. Under these conditions, we find our value of n increases from 2.7 to 7.0, indicating that our reported results are a conservative estimate.

3) The reviewers wanted more comparisons between our approach and results to other papers in the field. Towards this end, they suggested several excellent papers to include. We have incorporated these and other references in our revised submission. These helped to strengthen our paper and give the reader a broader lay of the field by enabling more points of comparison in both the setup of our approach and discussion of the results. We appreciate that they took the time share specific, constructive feedback, which allowed us to easily understand and directly address their concerns.

Essential revisions:1) The Abstract and Introduction should be reshaped to reflect more of the background of CARs and T cell ligand discrimination and what a simplified signaling model with light based kinetic control could teach us.

In motivating our experimental approach, we have more heavily emphasized the advantages of a synthetic, CAR-based input. The simple, modular design of CARs facilitates easier engineering with our optogenetic approach and allows us to bypass many of the mechanisms the endogenous TCR might use to discriminate ligands. The fact that we observe kinetic proofreading with the more minimal CAR places constraints on the possible receptor mechanisms that contribute to sensing the duration of ligand binding. By directly controlling ligand binding half-life with a simple receptor, paired with live cell readout at different levels of the pathway, we are able to better isolate and study the extent to which the proximal signaling pathway aids in the kinetic discrimination of ligands.

2) Many aspects of the system are poorly described- at this point in the technologies they are working with they should provide:

We agree that a more thorough quantitative evaluation of our system would be helpful for readers to interpret our results and would also aid in replication of our work by others. We have addressed all of the experimental concerns below.

a) Absolute quantification (molecules/cell) of the CAR on the Jurkat cells. This could be done by flow cytometry with 488 or 640 labelled Luv2 (dark) and MESF beads from Bangs labs.

We used beads with known antibody binding capacity (Bang’s lab Simply Cellular beads) to measure the absolute number of CARs on each cell line. We also measured TCR levels on wt Jurkats for comparison. We find that there are between 62-238 x10^3^ CAR molecules on the surface of our cells lines, compared to 273-547 x10^3^ TCR molecules on wt Jurkats. The quantification and controls are detailed in Figure 1—figure supplement 4.

b) The density of the Lov2 domain in the SLB in molecules/µm2 across the range that they are using. This could be done by line scanning FCS (dilute fluorescent version with non-fluorescent version if density is too high.) See PMID: 19254560.

In lieu of a line scanning FCS, we used single molecule TIRF microscopy to measure the absolute density of LOV2 on the bilayer. Fluorescent LOV2 was diluted into non-fluorescent LOV2 and added to the bilayer at saturating concentrations. By counting the single molecules, we calculated the total LOV2 binding capacity of the bilayer at saturation. We then measured what fraction of saturation our experimental conditions achieved and used this value to calculate the absolute concentration of LOV2 on the bilayers in our experiments.

We find that over the range of LOV2 concentrations we used, there are between 40 and 1200 molecules/um^2^. This is in a similar range to what others have reported with a CAR on planar lipid bilayers (between 0.1-800 molecules/um^2^; Taylor et al., 2017).

c) The estimate of occupancy will be inaccurate if they don't include a control for exclusion of non-binding Luv2 mutant or similar sized protein in the bilayer with a difference fluorophore. See: PMID: 17911103.

It is true that exclusion of non-bound LOV2 from the synapse could affect our receptor occupancy measurements. To assess how this might affect our results, we recalculated our results under a worst-case scenario: all unbound LOV2 molecules are excluded from synapse. We found that this only increased the magnitude of observed kinetic proofreading from a value of *n* of 2.7 to 7.0. Thus, we believe we report a conservative estimate of the degree of proofreading.

3) It would be useful to see at least diffraction limited higher magnification images of the different signals to established the correlation between IRM contact, CAR occupancy, the ZAP-70 and C2 domain sensor. It the CAR forming microclusters or is forming larger domains as it appears in the zoomed out views provided.

We have included higher magnification images of CAR occupancy, reporter localization and IRM contact for both the ZAP70 and DAG biosensors (Figure 1—figure supplement 3). ZAP70 shows a clear co-localization with sites of receptor occupancy, as expected. DAG has a diffuse localization throughout the synapse. Both the sites of receptor occupancy and reporter activity are contained within the IRM footprint. The CAR does appear to form larger domains when bound to LOV2. However, we cannot rule out that single CAR molecules are also occupied and signal, as background free LOV2 prevents single molecule imaging.

4) A number of relevant papers need to be cited that are pertinent, and the present results related to these studies. A small sampling of the missing literature are:

Overall, these papers are excellent and will help us more fully discuss the context of our work. Below, we briefly describe how we discussed each of the suggested works and provide their location in the revised text.

a) The work from the Davis lab from several years ago on using caged pMHCs that were sensitive to light, which allowed measurements that described the speed at which signals propagate downstream.

The speed of onset to DAG production in our system is comparable to the ~8 second “offset time” described in this work. While we both use optical methods of stimulation, we cannot directly measure such an “offset time” because blue-light acutely terminates signaling in our system. Once blue light is removed, LOV2 goes through a slow reversion to an active state with a half-life of ~30 seconds. That DAG levels reach their maximum on this timescale suggests that once a CAR is bound, DAG can be produced within seconds. (Subsection “LOV2 binding induces cell signaling”).

b) The role of rebinding that allows ligands with short half-lives to signal, described by work from eight years ago by Van der Merwe and co-workers (Immunity) and Huseby and co-workers.

This work is important because it shows that sufficiently fast on-rates can effectively occupy receptors longer than their binding half-life. Not only does this highlight the importance that other binding kinetics can play in ligand discrimination, it also allows us to point out that the 4-7 second transition between stimulatory and non-stimulatory half-lives we observe may in fact be an underestimate. The immobilization time the CAR experiences may be longer if LOV2 rebinding is sufficiently fast. (Subsection “LOV2 binding induces cell signaling”).

c) The role of co-receptors, amount of active Lck and diffusion in kinetic proof reading described by Stepanek et al., (2014).

This work made a great discussion point to contrast our results with, given that we don’t observe kinetic proofreading effects at the level of ZAP70 recruitment. We used it to remind readers that our system does not engage co-receptors and so may be blind to additional kinetic proofreading steps in the endogenous TCR. (Discussion section).

d) You also ignore the data the Germain lab, and others, have presented showing that the same ab TCR signals differently in DP thymocytes vs. mature cells [in particular, allows meaningful propagation of signals for gene expression in DP with pMHC ligands incapable of electing the same response or biochemical changes in a mature T cell], meaning that intracellular events can control signal propagation separately from occupancy, half-life or mechanosensing.

Since our system specifically manipulates ligand binding half-life, we tended to focus our attention on how binding half-life affects T cell activation. This work reminded us to emphasize that the T cell response is not entirely dictated by a ligand’s properties, but also is sensitive to the internal state of the T cell. (Subsection “Quantifying the degree of kinetic proofreading”).

5) The model for proofreading is a bit strange. Many people have made models that are far more realistic and incorporate feedback, including several papers from the Germain lab (1995; 1997; 2005; 2002), Chakraborty and Palmer, Chakraborty and Weiss, and the more theoretical efforts by Murugan, Huse, and Leibler. These papers should at least be cited and briefly discussed.

We decided to use a simple model of kinetic proofreading to reduce the number of free parameters and provide a conservative estimate of the degree of proofreading. However, as noted, the trade-off is that the model does not have direct analogy to the numerous biochemical steps that others have shown to be important in ligand discrimination. Thus, our model differs from the data in some important ways, most notably that our data shows cells are less sensitive to receptor occupancy than the model predicts. We now explicitly discuss these limitations of our model and how it could be improved by incorporating aspects of previously published models. (Subsection “Quantifying the degree of kinetic proofreading”) and elsewhere throughout the paper.

6) The estimate of 3 proofreading steps is undoubtedly wrong. The various steps do not have the same time scales and there are feedback loops. These effects make the model's fit to obtain n very difficult to interpret. This part needs to be deleted.

We agree that 3 proofreading steps is likely an oversimplification and may be confusing to some readers. The corresponding section has been removed.

7) The comments about ZAP70 not being subject to kinetic proof reading in the Introduction need to be made with great care. Two points to note in this regard: a] There are data suggesting that partial phosphorylation of ITAMs is at least correlated if not connected to antagonism. b] in vivo, ZAP that is bound to TCR is not phosphorylated. Also note that in vivo the T cells are constantly stimulated by self ligands (see recent paper by Weiss, Chakraborty and co-workers in MCB).

We think that the reviewer is raising the important point that equivalent amounts of ZAP70 recruitment do not necessarily mean equivalent amounts of receptor phosphorylation or ZAP70 activation. We are more careful in the current manuscript to restrict our claims to our direct experimental measurements—kinetic proofreading is not observed at the level of ZAP70 recruitment but is observed at the level of DAG accumulation.

[Editors' note: further revisions were requested prior to acceptance, as described below.]

The reviewing editor has asked that you address one point prior to a final decision. Please describe the steps in optimisation of the adhesion system and the specific requirement for the system chosen. Also, the implications of this adhesion system for kinetic proofreading – aka kinetic segregation model – should be discussed. For example, could the lack of kinetic proofreading in TCR proximal signaling be due to the presence of the adhesion system, which takes care of CD45 exclusion and makes signaling a matter of occupancy? If the light dependent kinetics were in control of CD45 segregation, would this step show a sensitivity to kinetics?Recent relevant papers include:Chang et al., 2016.Cai et al., 2018.The editor is not asking for additional experiments, just a transparent discussion of the results of the adhesion system, why anti-β2m, and if this is actually required for the signaling by the light inactivated CAR.Reviewer #1:The study continues to struggle with proving "kinetic proofreading" (KP) due to the issue that the light dependent acceleration of dissociation of the LOV2 domain from Zdk1 doesn't impact the dark reversion rate (on rate) so the affinity is decreased by blue light. This is a similar limitation faced by people working with mutation-based systems for TCR or CARs. The mass action equation makes it very hard to change the half-life while maintaining a constant affinity and occupation. They partly compensate for this by doing the experiments at different densities of the LOV2 domain to have matched receptor occupancy at all light intensities. These densities are now specified, but there is a surprise there. This is partly successful as they do see a better correlation of DAG to kinetics rather than occupation, but it’s expected that this system fires fully at low occupancy, so this lack of linear relationship to occupation is expected. Thus, the proof of KP is not advanced that much in practice, but the potential is there. Additional innovations seem necessary to fully exploit it.Essential Revisions:The authors have added a synthetic adhesion system, which was only mentioned in the methods and some legends. They use a biotinylated mAb to β2m, which should pull the cells very close to the substrate, perhaps less than 10 nm. I assume this will exclude CD45 at least partially if not quite robustly (the authors should check this) so all they need to do to trigger the CAR is pull them into these close contacts. This changes everything as they are not making the LOV2-Zdk1 interaction achieve KP through "kinetic-segregation". This is thought by many to be a critical process that needs to be mediated by the ITAM bearing receptor to initiate signalling in many contexts. So, this adhesion system and its implications need to be described in the experimental design and schematics in figures as its likely critical to interpretation. Can the system work at all without this synthetic adhesion system? This may explain why they see no kinetic proofreading at the level of ZAP-70- as its recruitment is taken care of by the adhesion system and then bringing the CAR into the close contacts.The other surprise is that the binding of the LOV2 domain displays significant positive cooperativity in binding to the SLB, suggesting that it oligomerises. How does this impact the measurements and interpretation?

We initially included the passive adhesion system to ensure that a decrease in the TIRF signal could not be due to the cells detaching from the bilayer. Ultimately, this system is dispensable, as experiments included in the supplement (Figure 4—figure supplement 3) without the adhesion system show an equivalent degree of kinetic proofreading (*n* in our model) at the level of DAG. Thus, the passive adhesion system was not used for the ZAP70 data and could not impact those results. However, we point out that the kinetic segregation of CD45 may impact kinetic proofreading differently in the native TCR-pMHC context, where the size of the extodomains may differ from our CAR-LOV2 pair. Revisions to reflect these changes in the Discussion section. Additionally, we added two more references of recent papers that highlight kinetic proofreading outside of the TCR as examples of areas where our approach may have future utility.

[Editors' note: further revisions were requested prior to acceptance, as described below.]

Thank you for submitting your article "Light-based tuning of ligand half-life supports kinetic proofreading model of T cell signaling" for consideration by eLife. Your revised article and response to our request has been reviewed by a Reviewing Editor and Arup Chakraborty as the Senior Editor.Unfortunately, your response has not addressed the only issue that needs to be resolved. When the question about adhesion first came up, we asked about low affinity interaction in the off state that might support residual adhesion. You noted that this was not a relevant question because the adhesion system was added to make the system independent of the CAR interactions with the ligand to keep the cells in place over many cycles. Now you say the adhesion system is not needed and in fact you did not use it for the studies looking at ZAP-70's role in kinetic proofreading. Given this, unfortunately, we need to ask again if this indicates that there is a low affinity interaction in the 100% "off" state that could mediate some residual adhesion, but without signaling. If the cells remain attached in the presence of the most intense blue light that would completely turn off the interaction, then it seems there would be support for the presence of such an interaction. Can you acknowledge this, or do you have another explanation for the system working with or without the parallel adhesion system?

For the ZAP70 experiments where passive adhesion was not used, we agree with the reviewer that the residual adhesion could be due to many weak interactions with LOV2 in the low-affinity state. An alternative possibility is that because saturating amounts of blue light (as assayed from our in vitroLOV2-Zdk binding experiments) were not needed to maximally suppress cell signaling, cells were never exposed to saturating amounts of blue light, meaning a small fraction of LOV2 likely still existed in the high-affinity conformation. This small population of LOV2 may also account for the residual adhesion we observed. Both of these possibilities are now discussed to provide a more complete technical description of the new system.